

# Strato-mesospheric carbon monoxide profiles above Kiruna since 2008

Niall J. Ryan[1], Mathias Palm[1], Uwe Raffalski[2], Richard Larsson[3], Gloria Manney[4], Luis Millán[5], Justus Notholt[1]

[1]Institute of Environmental Physics, University of Bremen, Bremen, 28359, Germany
[2]Swedish Institute of Space Physics, Box 812, SE-981 28 Kiruna, Sweden
[3]National Institute of Information and Communications Technology, Tokyo, Japan
[4]NorthWest Research Associates, Socorro, New Mexico, USA and Department of Physics, New Mexico Institute of Mining and Technology, Socorro, New Mexico 87801, USA
[5]Jet Propulsion Laboratory, M/S 183-701, 4800 Oak Grove Drive, Pasadena, CA 91109, USA

*Correspondence to*: Niall J. Ryan (ryan.niall@gmail.com)

**Abstract.** This paper presents the retrieval and validation of a self-consistent timeseries of carbon monoxide (CO) above Kiruna using measurements from the Kiruna Microwave Radiometer (KIMRA). The spectra are inverted using an optimal estimation method to retrieve altitude profiles of CO concentrations in the atmosphere within approximately 48 - 84 km altitude. Atmospheric temperature data from the Special Sensor Microwave Imager/Sounder aboard the US Air Force meteorological satellite, DMSP-F18, are used in the inversion of KIMRA spectra between January 2011 and May 2014. This dataset is compared with CO data from Microwave Limb Sounder aboard the Aura satellite and shows a high level of agreement at all altitudes: There is a maximum bias for KIMRA of ~ 0.65 ppm at 68 km (corresponding to 14.7% of the mean CO value at 68 km), and correlations between the instruments are within 0.87 and 0.94. To expand the CO dataset outside of the lifetime of DMSP-F18, another inversion setup was used that incorporates modelled temperatures from the European Centre for Medium-Range Weather Forecasts. The effect on the retrieved CO profiles when using a different temperature dataset in the inversion was assessed. A comparison of the two overlapping KIMRA CO datasets shows a bias of < 5% and a correlation > 0.98 at all altitudes below 82.5 km. The extended dataset shows a higher variation (≤ 6%) in CO concentrations that is not explained by random error estimates. The extended KIMRA CO timeseries currently spans 2008 to 2015, with gaps corresponding to non-operation and summer periods when CO concentrations below ~ 90 km drop to very low values.

The data can be accessed at: https://doi.pangaea.de/10.1594/PANGAEA.861730.

## 1 Introduction

The principle source of carbon monoxide (CO) in the middle atmosphere is the photolysis of carbon dioxide ($CO_2$) in the thermosphere and its subsequent vertical transport, and its only sink is through reaction with the hydroxyl radical ($\cdot$OH)



(Solomon et al., 1985). The loss rates in the thermosphere are low and this leads to a strong vertical gradient in CO concentrations. As the production and loss mechanisms for atmospheric CO require the presence of sunlight, the lifetime of CO during polar winter is on the order of months (Solomon et al., 1985, Allen et al., 1999), making it an excellent tracer for atmospheric dynamics. In spring the lifetime in the upper stratosphere can be 15 – 20 days poleward of 60˚ latitude

(Minschwaner et al., 2010). Due to the longer CO lifetimes within the polar vortex during winter, there exists also a strong horizontal concentration gradient across the vortex boundary.

While satellite measurements of CO profiles have been used regularly to study atmospheric transport processes, particularly during northern winters (e.g. Damiani et al., 2014, Lee et al., 2011, Manney et al., 2009, McLandress at al., 2013), ground-based CO profile datasets for the poles are few and far between. The Ground-Based Millimeter-Wave Spectrometer (GBMS)

installed in Thule Air Base, Greenland (76.5°N, 68.7°W), has been used to study the composition of Arctic winter of 2001/2002 (Muscari et al., 2007) and the Sudden Stratospheric Warming (SSW) of 2009 (Biagio., et al., 2010). The Onsala Space Observatory instrument (57°N, 12°E) has measured CO in 2002-2008 (Forkman et al., 2012), and from 2014. The British Antarctic Survey (BAS) Radiometer dataset for Troll Station (72°S, 2.5°E) covers February 2008 to January 2010 (Straub et al., 2013). Each of these three ground-based instruments are microwave radiometers that measure emissions from

molecules undergoing rotational transitions in the atmosphere, offering the advantage of providing measurements during polar night.

This paper presents a CO profile dataset from 2008 to 2015 from measurements made by the Kiruna Microwave Radiometer (KIMRA) at the Swedish Institute for space Physics, Kiruna (67.84°N, 20.41°E). KIMRA measurements during the winters 2008/2009 and 2009/2010 have previously been used (Hoffmann et al., 2011) to retrieve CO profiles that have been

compared to satellite data from: the Microwave Limb Sounder (MLS) aboard Aura, the Atmopsheric Chemistry Experiment - Fourier Transform Spectrometer (ACE-FTS) aboard SCISAT-1, and the Michelson Interferometer for Passive Atmospheric Sounding (MIPAS) aboard ENVISAT. The comparisons showed good agreement below 60 km but at higher altitudes the profiles significantly diverged leading to a bias of > 5 ppm at 80 km. The shape of the bias in the profile was consistent between satellite datasets and its origin is unclear.

This paper presents a new CO dataset from KIMRA measurements, beginning in 2008, and retrieved using a new inversion setup. The layout of the paper is as follows: Section 2 provides details on the KIMRA observation system: the measurements and the inversion technique. Section 3, to establish validity of the observation system, presents a comparison of the KIMRA data with MLS data, using temperature information from the Special Sensor Microwave Imager/Sounder (SSMIS) (Kunkee et al.,2008; Swadely et al., 2008) aboard the US Air Force's Defense Meteorological Satellite Program F-18 satellite as input

for the KIMRA CO inversion. Section 4 investigates changes in the retrieved KIMRA CO data when using temperature information from the European Centre for Medium-Range Weather Forecasts (ECMWF), and presents this extended dataset, currently spanning 2008-2015. Satellite data is exchanged for ECMWF analysis in the KIMRA inversion in order to have a consistent temperature input for the entire KIMRA dataset; The timespan of continuing KIMRA measurements will generally surpass the lifetime of any satellite instrument. Section 6 offers some concluding remarks.



## 2 Instrument and dataset

### 2.1 KIMRA

KIMRA is housed at the Swedish Institute for Space Physics (IRF), Kiruna, and was partly designed by the Institute for Meteorology and Climate Research (IMK) at the Karlsruhe Institute of Technology (KIT) (Raffalski et al., 2002). KIMRA

utilises the frequency range of 195 – 233 GHz and has been measuring, among others, atmospheric spectra that correspond to the J = 2 → 1 rotational transition (230.54 GHz) of CO. KIMRA has operated in Kiruna since 2002 and has been making measurements of CO emissions since 2008. The more general aspects of the instrument are given in Table 1 and the more specific details can be found in Raffalski et al. (2002) and Hoffmann et al. (2011). The spectrometer used for CO measurements is a Fast Fourier Transform Spectrometer (FFTS) made by Omnisys Instruments, with 1024 channels and

used with a bandwidth of 110 MHz to give a resolution of ~107 kHz per channel. For a measurement, KIMRA points to the atmosphere at an angle that is chosen to provide the best signal-to-noise (SNR) at that time, which is governed by the atmospheric conditions. So the pointing angle changes from one measurement to another, meaning that individual spectra cannot be averaged to reduce the SNR. Rather, each spectrum is used to retrieve a CO profile, which may then be used in an average. Figure 1 (left) shows the distribution of the duration times of each measurement.

### 2.2 Inversion setup

CO profiles are retrieved from the spectra using an optimal estimation inversion (Rodgers, 2000). This is a Bayesian statistical approach that constrains the retrieved CO profile according to some a priori atmospheric information. The inversion was carried out using the Qpack 2 (Eriksson et al., 2005) package, which employs the Atmospheric Radiative Transfer Simulator: ARTS 2 (Eriksson et al., 2011) to model radiative transfer through the atmosphere, i.e., the forward

model. All of the following information that is input into the inversion is done so using Qpack 2. The a priori CO information used here is the average of one winter of output from the Whole Atmosphere Community Climate Model (WACCM) (Garcia et al., 2007) provided by Douglas Kinnison at the National Center for Atmospheric Research (NCAR), with a standard deviation of 100% at all altitudes. Ozone ($O_3$) is also retrieved simultaneously with CO, as there is an $O_3$ spectral line located at 231.28 GHz, and attenuation of the CO spectral line due to water vapour is accounted for by

including the water vapour continuum described by Rosenkranz (1998) in the forward model and inversion. The spectroscopic information used is from the HITRAN 2008 catalogue (Rothmann et al., 2009). Continua of molecular oxygen ($O_2$) and nitrogen ($N_2$) (Rosenkranz, 1993), and nitric acid ($HNO_3$) lines are also included in the inversion but are not retrieved and are considered model parameters.

Measurement noise (statistical noise on the spectrum) was estimated by fitting a second order polynomial to a wing of the

spectrum and calculating the standard deviation of the fit. As there is no windowing applied in the operation of the FFTS, the spectrometer channels are specified in the inversion as having a sinc-squared response function (Harris, 1978). Three sine wave functions are fitted to the baseline of each spectrum during an inversion to account for errors in the baseline, which are



most often produced by standing waves in the instrument. The periods of the sine waves were found by first inverting the spectra without a fit to the baseline and then evaluating a periodogram of the residuals. This procedure was also applied to subsets of the data in case some changes in the baseline with time became evident. For an inversion, the periods of the sine waves are fixed in frequency space and the amplitudes and phases are retrieved. The periods of the fitted waves are 27.5 MHz, 55 MHz, and 36.3 MHz, with an estimated uncertainty in the amplitudes of 0.5 K, 0.3 K, and 0.5 K, respectively. A second order polynomial is also fitted to the baseline to account for any offsets or long-period sine wave signatures; The zeroth, first, and second order coefficients have estimated uncertainties of 1 K, 0.5 K, and 0.5 K, respectively.

The altitude, pressure, and temperature information (zpT) for the inversion is constructed in two ways: For the first case, used in the comparison work with MLS (January 2011 to May 2014): information up to 10 hPa (~ 30 km) is from daily National Centres for Environmental Prediction (NCEP) profiles; information for 10–0.01 hPa (~ 30–80 km: recommended range for use) is from SSMIS; information above that is from the NRLMSISE-00 empirical model of the atmosphere (from herein called MSIS) (Picone et al., 2002). Temperature data from SSMIS currently begins in January 2011 and ends in June 2014. For the second case, used in the temporal extension of the KIMRA CO dataset (2008 to 2015): the information up to 0.01 hPa is from ECMWF Operational Analyses output and information above that is from MSIS. The SSMIS dataset was used because it compares well with other satellite datasets (personal communication with Richard Larsson, and Patrick Sheese at University of Toronto) and has approximately four colocations (sets of measurements within 66–68 ˚N, 15–25 ˚E) with Kiruna per day. Around the altitudes at which the different temperature profiles are merged for use in the KIMRA inversion, the data are smoothed to avoid discontinuities in the final temperature profile. The inversions utilising SSMIS data are considered as those using the most suitable available data for the CO inversion, as the KIMRA CO profiles are most sensitive to temperature information between approximately 40 and 90 km, and the resulting CO dataset is considered as a reference point for inversion setups using alternate input temperature information.

The pressure grid used in the forward model is 250 layers, spaced approximately equally in altitude, between the ground and 125 km. The retrieval grid is a 62-layer subset of the forward model pressure grid with approximately 2 km spacing between the ground and 124 km. Using a subset of the forward model grid is recommended by Patrick Eriksson (first author of Qpack 2) as giving the most accurate mapping of information from the forward model grid to the retrieval grid during an inversion. An estimate of the errors in the CO profiles due to model and instrumental parameters can be found in Hoffmann et al. (2011).

## 2.3 Characteristics of the retrieved dataset

This section discusses the profiles retrieved using the NCEP/SSMIS/MSIS temperature information, between 2011 and mid-2014. The inverted CO dataset is restricted to the months of September to May, as summertime CO concentrations in the middle atmosphere are very low. The data are then further filtered to those that satisfy the following: a converged inversion, a degree of freedom for signal (DOFS) greater than 1 (DOFS are calculated as the trace of the averaging kernel matrix (Rogers, 2000)), a standard deviation of the fit residual no greater than 1.5 times the initial estimate of the measurement



noise (to avoid overfitting the measurement), a mean of the fit residual that lies in the range (-1 K, 1 K), and a baseline brightness temperature of < 230 K (an ad hoc indication of too cloudy weather). No filtering for outlying or anomalous concentration values was applied to the data. 28% of the data was identified as unusable, with the DOFS criteria being responsible for about half of that number. If one finds this a high rejection rate, bear in mind that KIMRA operates

regardless of weather conditions, and CO concentrations may be very low in time periods near the beginning and end of winter.

Figure 1 (middle) shows an example spectrum of a CO measurement with KIMRA and the corresponding fit from the inversion. The mean of the averaging kernels for the CO dataset are also shown in Figure 1 (right) along with the measurement response (sum of the rows of the averaging kernel matrix) and the altitude resolution of the profiles (full-

width-at-half-maximum of the averaging kernels). Altitudes with a measurement response greater than 0.8 are considered to represent the range of useful profile information: the retrievable altitude range. This choice is somewhat arbitrary, with 0.8 being a regularly used value (e.g., Forkman et al., 2012; Hoffmann et al., 2011; Straub et al., 2013). The CO profiles here have an average retrievable altitude range of 48–84 km and a vertical resolution of between 15 and 18.5 km depending on the altitude. These compare favourably to an altitude range of 40-80 km and resolution of 16-22 km for the CO profiles

shown in Hoffmann et al. (2011) for the winters of 08/09 and 09/10. The peaks of the averaging kernels, when represented in volume mixing ratio, are shifted down in altitude compared to a representation using relative concentrations. Hoffmann et al. (2011) provides a detailed discussion of the representation of averaging kernels for ground-based CO measurements. The DOFS for the current dataset have a mean of 2.0 and a standard deviation of 0.6. When comparing KIMRA CO profiles to those of an instrument/model that has a significantly different vertical resolution, the full KIMRA averaging kernel matrices

and a priori should be used to smooth (Rodgers and Connor, 2013) the other profiles.

## 3 Comparison with MLS

This section presents a comparison of CO profiles from KIMRA and MLS that are colocated in space and time. A description of the MLS instrument is given in Walters et al. (2006). Version 4.2 of the CO data (Schwartz et al., 2015) is used here, a description of which can be found in Livesey et al. (2015). Version 4.2 CO data covers the pressure range of

215-0.0046 hPa and has an altitude resolution between 3.8 and 6.2 km. The precision of the CO profile reaches a maximum (largest) value of 1.1 ppmv at the highest retrieval layer (0.0046 hPa). In the middle atmosphere the dataset has a positive bias of approximately 20% compared with the ACE satellite, suggested by Livesey et al. (2015) using a validation of the MLS version 2.2 CO data (Pumphrey et al., 2007).

### 3.1 Colocation of KIMRA and MLS measurements

For a given KIMRA measurement, a coincident MLS measurement was defined as follows. MLS measurements that were made within 4 hours, ±2° latitude and ±10° longitude of the KIMRA measurement, and lie in the same position relative to



the vortex edge as the KIMRA measurement (inside, outside, or within the edge of the polar vortex) were identified. The MLS measurement from this group that was closest, along a great circle, to the KIMRA measurement was chosen as coincident. A given KIMRA/MLS measurement could only be considered coincident with one MLS/KIMRA measurement. The location of a measurement with respect to the polar vortex was determined using scaled potential vorticity (sPV) values

from NASA's Global Modeling and Assimilation Office's (GMAO's) MERRA (Modern Era Retrospective analysis for Research and Applications) (Rienecker et al., 2011). The sPV values for KIMRA were calculated geometrically along the instrument's line of sight. sPV values of 1.6 and 1.2 x$10^{-4}$ s$^{-1}$ have been used extensively in previous works (e.g. Manney et al., 1994, 2007, 2011; Jin et al., 2006) to define the respective inner and outer edge of the vortex, and the same values are used here.

The position relative to the vortex and the distance between measurements was calculated at 50 km altitude. The position relative to the vortex changes with altitude, which means that a single profile may simultaneously contain information from inside/outside/the edge of the polar vortex. 50 km was chosen as the altitude to define the three positions relative to the vortex because it divides the CO measurements from KIMRA into the three most distinct populations of concentration values. This was tested by using different altitudes to define the position relative to the vortex, calculating partial (40-60 km

and 60-80km) CO column concentrations, and testing whether the column concentrations were significantly different from each other when grouped by vortex relative position. This is not to say that that there should always be three distinct air masses of CO but it can be expected based on the strong cross-vortex gradient of the gas (see Section 1). The strong CO gradient during winter has recently been used in a chemical definition of the mesospheric vortex (Harvey et al., 2015).

Figure 2 shows the results for the KIMRA CO columns with vortex relative positions calculated using sPV values at 40, 50,

and 60 km. The sPV information is available up to approximately 62 km. Using 40 km, the partial columns defined as outside and in the edge of the vortex are statistically indistinguishable at the 5% significance level using an unpaired two-sample t-test, and using 60 km, the inside and edge of vortex 40-60 km partial columns are indistinguishable. With the same test and using 50 km, the three groups of partial columns comprise three distinct populations of concentration values in both the stratosphere and the mesosphere. Using other altitudes to define vortex relative position also produced distinguishable

concentrations and 50 km was chosen as it gives the most distinct vortex relative concentration values in both the stratosphere and mesosphere, and also because the sPV-defined location of the vortex edge in the upper stratosphere becomes much less well-defined as the winter progresses (Manney et al., 1997). Both the sPV and CO concentration gradients will be less distinct before/after and during the formation/breakdown of the polar vortex.

## 3.2 Comparison of colocated measurements

There are 916 coincident profiles found using the criteria in the previous section. Because the MLS profiles have a vertical resolution of more than twice that of the KIMRA profiles (Livesey et al., 2015), the MLS profiles were smoothed with the averaging kernels of coincident KIMRA profiles to account for the difference. MLS CO profiles are retrieved up to 0.001 hPa in the atmosphere and use a constant CO concentration above this. Because there is some low sensitivity of the



KIMRA CO profiles to atmospheric concentrations above this (see the averaging kernels in Figure 1), the MLS profiles were extrapolated from 0.001 hPa before smoothing. A linear extrapolation in pressure space was used to extend the MLS profiles instead of using scaled KIMRA a priori information so as to avoid creating artificial agreement between KIMRA and MLS. Figure 3 shows the mean of the extrapolated and the original MLS profiles, as well as the KIMRA a priori for comparison.

The extrapolated profile is considered a more realistic representation of the atmosphere than the constant value provided for MLS at these altitudes. MLS values at 82 km and 84 km are often defined as unusable for scientific work due to a too-high a priori contribution (Livesey et al., 2015) and the precision is given as negative in this case. Figure 4 (a-e) shows the results of the comparison of the 916 pairs of coincident CO profiles found between January 2011 and May 2014. The mean of KIMRA and MLS CO profiles are shown (a) as well as the mean of the differences (bias) in the coincident profiles (b), the

correlation between the profiles (c), and the slope of a line of best fit to KIMRA vs. MLS at each altitude layer (d), explained below. The times between coincident measurements and the location of the coincident MLS profiles are also shown (e). The statistics in Figure 4 are calculated using all colocated pairs of profiles and are also assumed to be representative of subsets of the data as there appears to be no particular time of the year during which the differences are more/less pronounced.

The mean profiles show good agreement with a maximum absolute bias of ~0.65 ppm at 68 km. The standard error on the

bias is not shown in Figure 4 (b) because it is very small due to the number of coincidences, but rather the standard deviation of the differences is shown as the whiskers on the bias. The combination defines a one-sigma space in which a KIMRA profile lies with respect to an MLS profile. The correlation of the profiles is high at all altitudes, only dropping below 0.90 above 82 km. Previous CO retrievals for winter 08/09 and 09/10 (Hoffmann et al., 2011) showed a bias with respect to MLS, increasing with altitude, with a value of > 5 ppm at 80 km. The structure of this bias was also shown in comparisons of

KIMRA with ACE and MIPAS, and it appears that this attribute is not present in the retrievals presented here.

The slope of a line of best fit to KIMRA vs. MLS measurements was calculated individually for each altitude layer as follows: for a given retrieval grid point the slope and intercept (or regression coefficients) for a line of best fit to the KIMRA and MLS values was calculated, accounting for errors in the X (MLS) and Y (KIMRA) values according to York et al. (2004). Two cases of KIMRA CO error estimates were used when calculating a line of best fit: the first being the

measurement error in the profile (the error due to statistical noise on the spectrum (Rogers, 2000)), and the second being twice the measurement error. The former is an underestimation of the true error on the profile as there are more error sources than measurement error alone, and the latter is likely an overestimation of the true error as the measurement error is a predominant source of error in the profile (Hoffmann et al., 2011). The results (Figure 4 (e)) show that the slope is always greater than the ideal value of 1.0 for both KIMRA error estimates, meaning that KIMRA shows a greater variation in CO

concentrations at all altitudes and the variation is not explained by the estimated random errors in the profile. The reason for the difference could be due to errors, e.g. spectroscopic information, calibration, baseline wave signatures, that can have a contribution that is neither truly systematic nor random. The difference in calculated slopes for the two KIMRA error estimates is insignificant within the standard error, likely due to the large natural variation in CO concentrations. Despite the



> 1.0 slope values, KIMRA and MLS are considered in good agreement due to the low bias and high correlation between profiles.

## 4 Extension of the KIMRA dataset

After establishing a reliable inversion scheme through comparison with MLS, the KIMRA dataset is extended in time by
substituting ECMWF Operational Analyses model output for the SSMIS temperature data in the inversion (see Section 2.2).

### 4.1 The effect of temperature input on KIMRA CO profiles

ECMWF temperatures are available four times per day, in six-hourly intervals beginning at midnight. The same filtering procedure for the retrieved data is employed as outlined in section 2.3. To evaluate the effect of using a different temperature dataset as input to the inversion, the two KIMRA datasets are compared where they overlap between January 2011 to May
2014 and the results are shown in Figures 5 and 6.

The mean and standard deviation of the differences between, the correlation of, and the slope of the lines of best fit for the two sets of temperature profiles (ECMWF/MSIS minus/vs. NCEP/SSMIS/MSIS) are shown in Figure 5. The same is shown for the two respective sets of CO profiles in Figure 6. The mean of the differences (bias) is also shown as a percentage to show small differences due to the change in temperature information. No smoothing with averaging kernels is applied to the
data.

The bias for the temperature profiles is very low below 50 km, showing good agreement between ECMWF and NCEP output, as well as the lower altitude SSMIS data, and then moves to a minimum of ~ -4% at 68 km. The maximum in the bias is about 4% at 118 km. The correlation is high below 50 km but has minima of < 0.50 at ~ 70 km and ~ 0.80 at 100 km. It should be noted that while MSIS is used in both temperature datasets, the time of the MSIS output is governed by the times
for the ECMWF output and the SSMIS measurements, and so the high-altitude (> 0.01 hPa) temperature values are not necessarily equal for the two inversion setups. The slopes of the lines of best fit were calculated without assuming uncertainty in the temperature datasets. The slope is within 10% of 1.0 below 46 km altitude, above which it decreases to around 0.50 for about 20 km, and then varies between about 1.0 and 0.7 at higher altitudes.

There is a general negative bias in the CO profiles that use ECMWF/MSIS, seen in (a) and (b) of Figure 6. The bias is small,
reaching a maximum of ~ 5% in the range of 68-78 km. The correlation of the profiles is very high, greater than 0.98 at all altitudes below 82.5 km. The slopes of the lines of best fit were calculated with the same error estimates described in Section 3.2. A value of 1.0 lies in the range of standard error of the slope below 56 km and above 80 km, and between these altitudes reaches a maximum of 1.06. As the only difference in the inversion setups is the temperature input, it follows that any inequalities of the respective KIMRA CO profiles are ultimately due to this difference. Overall, the CO profiles using the
differing temperature inputs shown here agree very well.





## 4.2 KIMRA CO dataset from 2008 to 2015

Figure 7 shows the KIMRA CO dataset between December 2008 and May 2015. Daily averaged CO concentrations between 46 and 86 km are shown. Data gaps in this time range are due to non-operation of the instrument or a lack of CO spectral line measurements. Data from winter 2015/2016 are presently unavailable due to a failure of the KIMRA cooling system.

While it is impossible to fully characterise the concentrations shown without inclusion of other instrument data and/or model output, some observations are made here. The beginning of each winter (from about September through November) shows a movement of higher CO concentrations to lower altitudes, which can in general be expected as predominantly due to vertical advection (Allen et al., 1999; Minschwaner et al.,2010; Solomon et al., 1985). The high CO concentrations remain for most of winter before decreasing again from March onwards, generally due to loss of CO because of increased ·OH and

movement of low-CO air from lower latitudes as the final warming of the pole occurs. Signatures of "major" SSWs, beginning 24[th] January, 26[th] January, and 6[th] January, 2009, 2010, and 2013, respectively, can be seen by the quickly (order of days) decreasing CO concentrations around these dates, and then the subsequent increases as the vortex recovered (Manney et al., 2009, 2015). The effects of a "minor" SSW in early January 2015 (Manney et al., 2015). can also be seen.

## 5. Data availability

The current KIMRA CO dataset (between December 2008 and May 2015), can be accessed publically through PANGAEA Data Publisher for Earth and Environmental Science at: https://doi.pangaea.de/10.1594/PANGAEA.861730. Metadata is also provided including the averaging kernel matrices for each measurement. It is recommended to use the averaging kernels specific to each CO profile when using the data in a comparison with model output or a dataset with significantly different altitude resolution.

## 6. Conclusion

The aim of this work was to create a self-consistent dataset of strato-mesospheric CO profiles above Kiruna, using measurements from the ground-based microwave radiometer, KIMRA, and an optimal estimation technique. The resulting profiles cover an altitude range of approximately 48-84 km. As a test of the validity of the KIMRA observing system and to create a reference dataset, CO was first retrieved using atmospheric temperature information from the SSMIS satellite

instrument, available within 2011-2014, and compared to CO data from MLS. The data agree well, with KIMRA showing a maximum bias of 0.65 ppm at ~ 68 km, and correlations with MLS greater than 0.90 at most altitudes. KIMRA shows a larger variation in atmospheric CO concentrations, compared to MLS, that is not explained by the estimates of random errors in the data, and may be due to some combination of random and systematic uncertainty. The KIMRA dataset is extended in time (2008 - 2015) by substituting ECMWF Operational Analyses temperature output in place of the SSMIS data. The

extended KIMRA dataset shows a difference (bias) of less than 5% compared to the reference dataset (the one using SSMIS



data), and correlations between the two are greater than 0.98 at most altitudes. There is a larger variation (≤ 6%) in the concentrations seen by the extended dataset, compared to the reference dataset over the same time period. The extended dataset currently spans the time between December 2008 to May 2015, with data gaps. Measurements are ongoing at IRF Kiruna.

## 5 Author contribution

M. Palm designed and took an active role in the project. N. Ryan developed the inversion setups for KIMRA and performed the comparisons. R. Larsson provided SSMIS temperature data. G. Manney and L. Millán provided sPV information for KIMRA measurements. U. Raffalski maintains KIMRA and provided the KIMRA measurement data. J. Notholt provided valuable feedback on the project. N. Ryan prepared the manuscript with contributions from co-authors.

## 10 Acknowledgements

This work has been funded by the German Federal Ministry of Education and Research (BMBF) through the research project: Role Of the Middle atmosphere in Climate (ROMIC), sub-project: ROMICCO, project number: 01LG1214A. We would like to express our gratitude to the MLS teams for making their CO and sPV products available. We would also like to thank the ECMWF and NCEP teams for making their products available, as well as the Qpack and ARTS communities for
15 making their software available. Work at the Jet Propulsion Laboratory, California Institute of Technology was done under contract with the National Aeronautics and Space Administration.

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



| System noise temperature | ~1800 K (single sideband) |
|---|---|
| Detector | Schottky diode at ~25 K |
| Sideband filter | Martin-Pupplett interferometer |
| Standing wave suppression | Path length modulator |
| Hot/Cold calibration | Blackbodies at ~195 K/~293 K |
| Spectrometer | FFTS. bandwidth/resolution: 110 MHz/ 107 kHz |

**Table 1: More general details of KIMRA at a glance. Also see Section 2.1.**





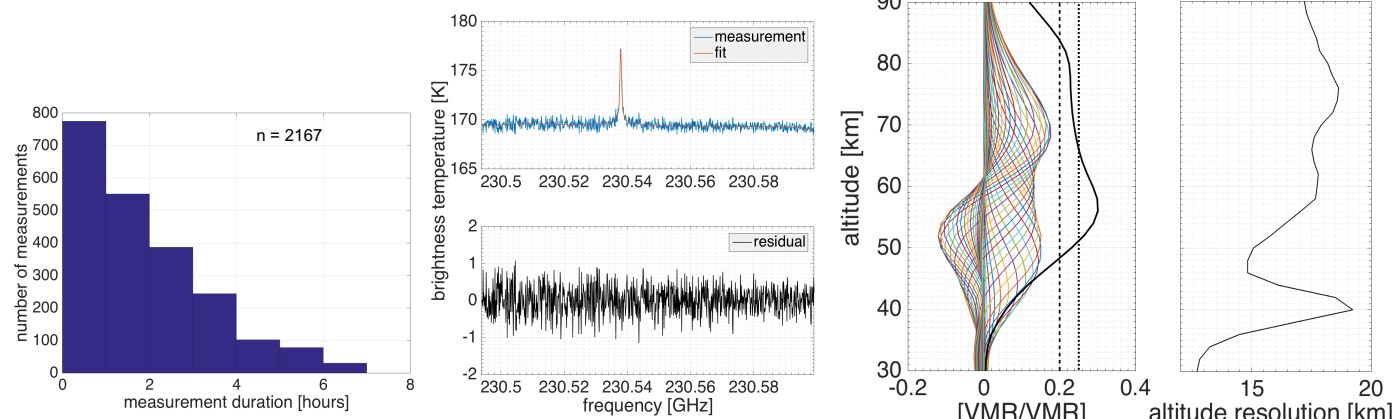

**Figure 1: Left: A histogram of the KIMRA CO measurement durations from January 2011 to May 2014 (see Section 2.2). Middle: (upper) An example measurement from November 5[th] 2012 with the corresponding fit from the inversion, and (lower) the residual of the fit. Right: (left) The mean averaging kernels for all CO measurements, with the measurement response divided by 4 shown in black. The dashed and dotted black lines indicate a measurement response of 0.8 and 1.0, respectively. (right) The corresponding mean altitude resolution of the CO profiles, derived from the FWHM of the averaging kernels.**



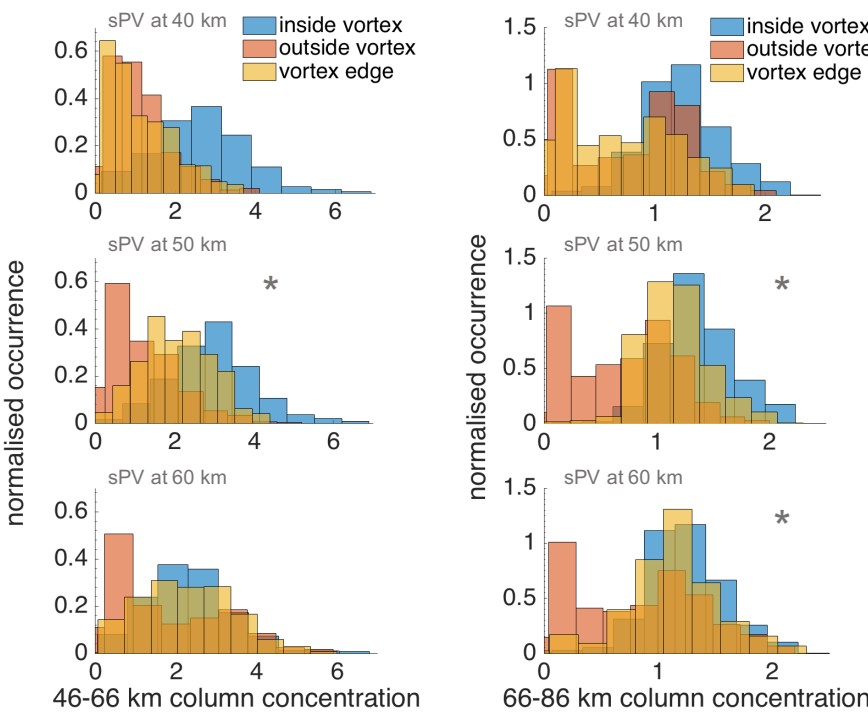

Figure 2: **The distributions of KIMRA CO partial (40-60 km and 60-80 km) column concentrations divided into groups defined by their position relative to the polar vortex edge using sPV (see Section 3.1). The relative positions are calculated using sPV values at 40, 50, and 60 km, as indicted on respective plots. Using sPV at 50 km gives the three most distinct distributions, as calculated with an unpaired two-sample t-test (See Section 3.1). The** ✱ **symbols indicate plots with three distinct distributions.**





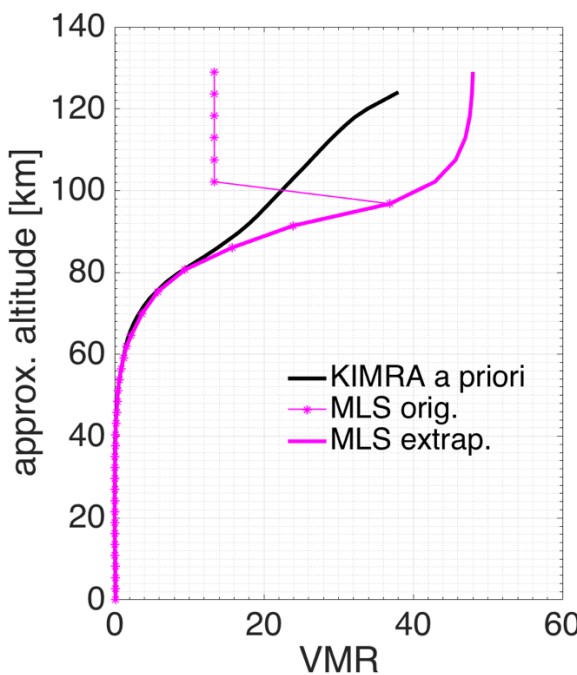

**Figure 3. The mean (in ppmv) of the coincident MLS CO profiles and the a priori CO profile used for the KIMRA inversion (Section 2.2). MLS orig. shows the mean of the supplied MLS profiles, which use a constant CO concentration above 0.001 hPa. The stars show the vertical location of the profile points. MLS extrap. shows the mean of the profiles that have been linearly extrapolated in pressure space above 0.001 hPa to provide more physical CO concentrations at these altitudes.**





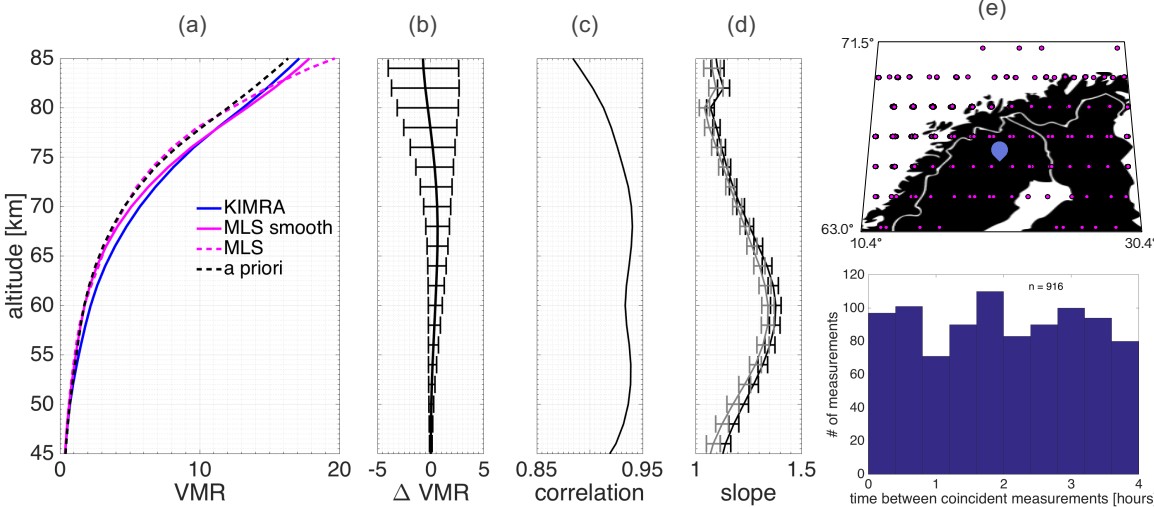

**Figure 4. (a) The mean of coincident KIMRA and MLS CO profiles (in ppmv) from 2011 to 2014, including the mean of the unsmoothed MLS profiles and the a priori profile used for the KIMRA inversion. (b) The mean of the difference between the**
5 **KIMRA and smoothed MLS profiles with the standard deviation of the differences as the whiskers on the line. (c) The correlation coefficients of KIMRA and smoothed MLS data. (d) The slope and standard error of a line of best fit to KIMRA vs. smoothed MLS, calculated at each level using given MLS error estimates and two estimations of KIMRA error: the measurement error (black) and double the measurement error (grey) (see Section 3.2). (e) The location of the MLS measurements (magenta) with respect to Kiruna (blue), and a histogram of the times between coincident measurements. The temperature input for the KIMRA**
10 **inversions shown here includes SSMIS data (see Section 2.2).**





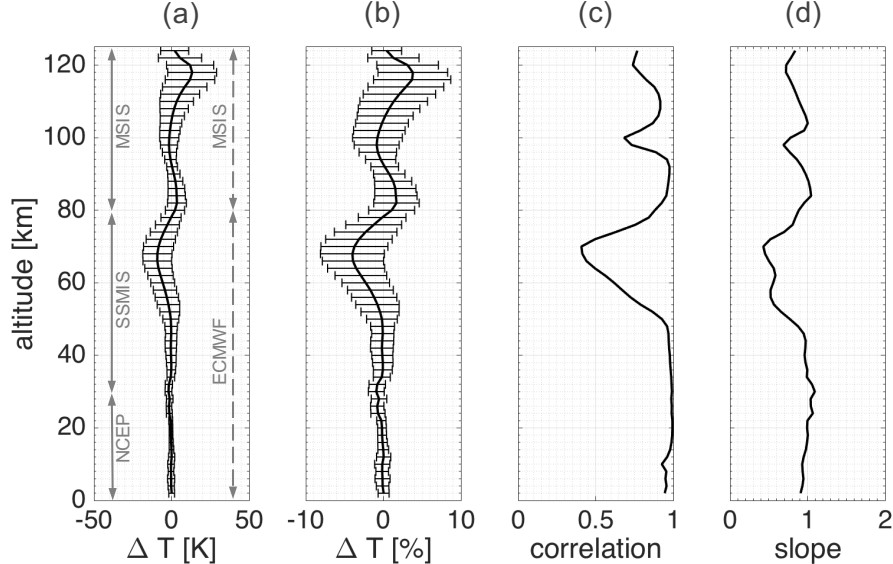

**Figure 5. (a) The absolute mean of the difference in the temperature profiles (ECMWF/MSIS minus NCEP/SSMIS/MSIS) used in the two inversion setups for KIMRA. The altitude ranges of the temperature information used in the profiles (see Section 2.2) are shown here. (b) The mean of the percentage difference in the profiles (difference divided by the average of the two profiles). (c) The correlation between the two datasets. (d) The slope of a line of best fit to the datasets at each level, with ECMWF/MSIS as the dependant (Y) variable.**


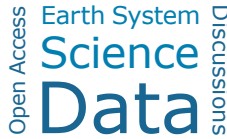

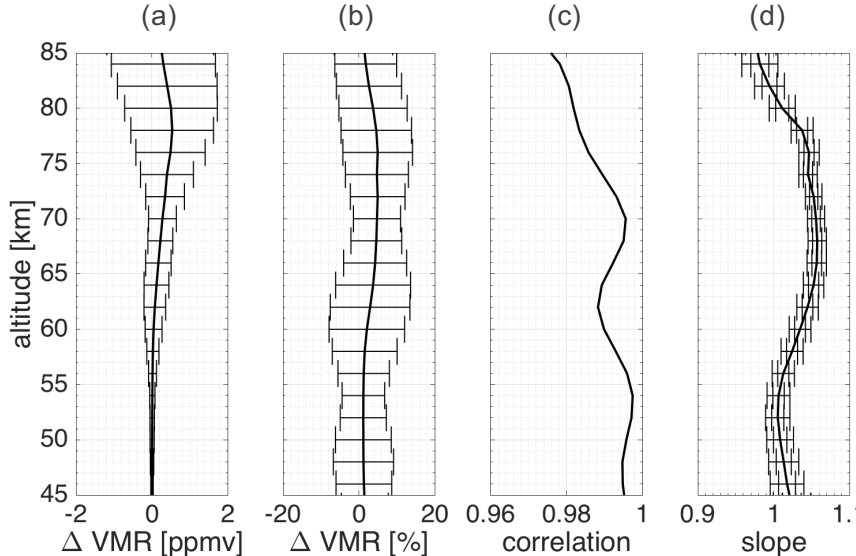

**Figure 6. (a-d) The same calculations as shown in Figure 5 but for the CO volume mixing ratios retrieved using the respective temperature datasets. The slopes and their standard errors (d) are calculated with the same two KIMRA error estimates as in Figure 4 (see Section 3.2), with the larger error bars corresponding to the larger error estimate. Note the different altitude range compared to Figure 5.**



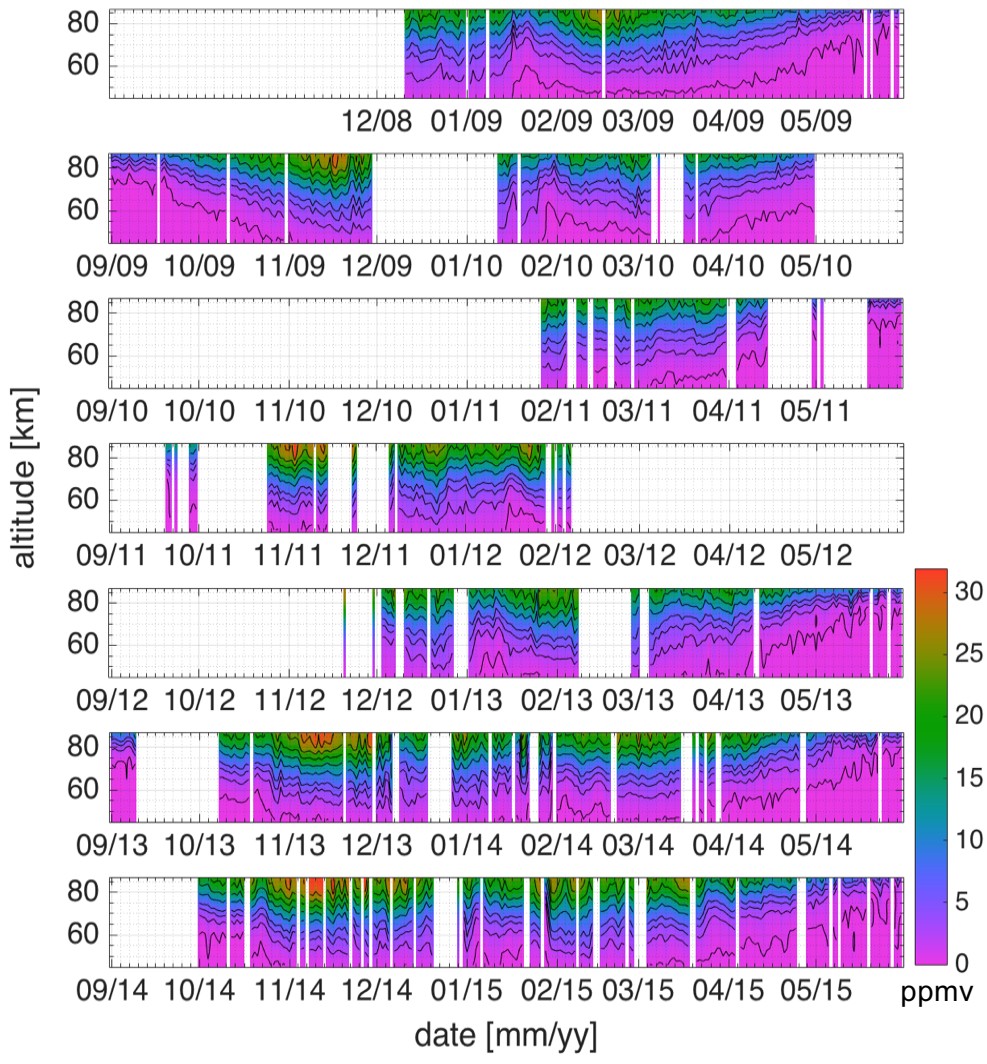

**Figure 7. Daily averaged CO volume mixing ratios (in ppmv) above Kiruna from December 2008 through May 2015. Blank areas within this time are gaps in the data record. Data is plotted using the Isoluminant colormap from Kindlmann et al. (2002), and non-uniformly spaced contours (black lines) between 0.4 and 28 ppmv are added to guide the eye. The temperature input for the KIMRA inversions shown here includes ECMWF analysis (see Section 2.2).**