# Peer review of "Strato-mesospheric carbon monoxide profiles above Kiruna since 2008"

_Earth System Science Data, 2016_

## Referee Comment (RC1) · Anonymous Referee #1 · 18 Oct 2016

General comments

The paper describes a new atmospheric dataset which comprises a time series of carbon monoxide (CO) vertical profiles above Kiruna, northern Sweden. It carefully details the derivation of CO volume mixing ratio profiles from ground based microwave radiometer measurements during 2008 to 2015. The methodology uses well-established atmospheric radiative transfer and retrieval codes and builds on previously reported work by the group on ozone profile measurements using the same radiometer instrument. The effect of a priori temperature on such retrievals can be important, and this is investigated using temperature profiles constructed from various observed and modelled (or reanalysis?) datasets. As well as describing how the profiles were determined and validated the paper indicates how they should be used e.g. applying the supplied averaging kernels to smooth other data-sets when making comparisons. Measurements such as these are important, providing insights into atmospheric dynamics and chemical transport in the polar middle atmosphere and having applications in climate modelling and 'ground-truthing' of satellite data.

Overall the paper is well structured and presented with adequate description of the methodology, discussion of the results, and citing of prior work. However some sections are somewhat challenging to follow, for example the discussion of differences between data-sets where it can be unclear which dataset is positively biased in comparisons with other data. I've highlighted a number of other areas in the text and figures where clarifications and improved presentation are needed. I recommend that the authors address these points before the revised paper is accepted for publication in Earth System Science Data.

Specific comments

Title. A date range of 2008–2015 would be better as 'since 2008' is ambiguous. It would be clearer if a more complete description of the measurement location was given in the title, e.g. Kiruna, Sweden including the geographic co ordinates.

Page 1, Line 15. 'CO concentrations' should be 'CO volume mixing ratio (VMR, in ppmv)' here and throughout the manuscript.

Page 2, Line 23. 'Leading to a bias of > 5 ppm at 80 km'. Here and throughout the paper it should be made clear what data the biases are between and which of the compared data-sets has the higher values. Also the units of VMR should be ppmv.

Page 2, Line 34–35. 'The timespan of continuing KIMRA measurements will generally surpass the lifetime of any satellite instrument'. This is an important point that could be expanded on, e.g. indicating what timespans / lifetimes are typically achieved by ground based and satellite instruments. What factors allow KIMRA, and other ground based instruments to make measurements that surpass those by individual, or overlapping, satellite missions. Why are long term datasets such as these important?

Page 3, Line 13. 'So the pointing angle changes from one measurement to another'. Does this mean that both the elevation and azimuthal angles are changed? If the azimuth is kept fixed while elevation is changed, as I suspect, then the azimuthal angle should be stated. Is the optimum pointing angle checked and, if necessary, changed after each calibration cycle measurement from which a brightness temperature spectrum can be calculated or after a longer time interval? What is the range of pointing angles?

Page 3, Lines 12–14. Please clarify whether 'individual spectra' and 'each spectrum' means those calculated from one calibration cycle, or integrated measurements over a longer period.

Page 3, Lines 20–23. Which version of WACCM is used, for which year is WACCM winter-time output used, and what are the WACCM grid points? If only one winter of WACCM data is used to generate the a priori CO profile is that representative for the years 2008–2015? Is an average profile for winter (presumably December, January, and February) appropriate for all of the months September to May where the CO profile will differ considerably?

Page 3, Lines 23–28. The source of the CO a priori data is given but not for the other atmospheric gases included in the forward model. How are the ozone, water vapour, molecular oxygen, nitrogen, and nitric acid a priori profiles constructed?

Page 3, Line 32. In this type of microwave measurement baseline errors include, and may be dominated by, those arising from standing waves. However there will be contributions to baseline error from imperfect radiometric calibration, non-linearity, and other instrumental artefacts introduced in the signal down-conversion and spectrometric analysis. While a detailed study of these errors, and their impact on the CO retrieval, is beyond the scope of this paper it would be helpful to indicate them here and later on in the discussion section. The contribution of baseline noise will also depend on the system noise temperature ($\sim$1800 K according to Table 1), spectrometer channel

bandwidth, and the integration time.

Page 4, Line 3. 'in case some changes in the baseline with time became evident.' Did any changes in the baseline become evident in processing this dataset?

Page 4, Line 5. How do the estimated uncertainties (0.5 K, 0.3 K, and 0.5 K) in the standing wave amplitudes compare with the baseline noise of the spectra? Are the standing waves being effectively fitted to the baseline noise level?

Page 4, Lines 19–20. Why are the KIMRA CO profiles most sensitive to temperature information between approximately 40 km and 90 km? Although the CO line profile will be dominated by thermal Doppler broadening in the upper mesosphere, at lower altitudes pressure broadening will dominate. This could also be discussed in the context of the averaging kernels of the CO retrieval and estimated altitude resolution (Figure 1 and Page 5, Lines 7–20).

Page 7, Line 6–7. MLS values at 82 km and 84 km are stated as often unusable for scientific work. Are MLS data with the warning flag at these altitudes included in the analysis and, if so, how might that affect the results?

Page 9, Line 4. Do you really mean the data from 2015/16 are presently unavailable? This suggests the data might become available in future, whereas my understanding of 'failure of the KIMRA cooling system' is that measurements were not collected during this period.

Page 9, Lines 10–13. What is happening to the CO during and after SSWs? What is the difference between major and minor SSWs and is the observed CO behave as would be expected for these events. What might be causing large decreases in CO abundance above ∼65 km during December 2013 (12/13 on Figure 7), that is not assigned to a SSW?

Technical comments

Page 1, Lines 17–18. 'This dataset is compared...' It should be made clear which

dataset is being referred to here.

Page 1, Line 24. 'at all altitudes below 82.5 km'. The exact altitude range should be given as this could be misinterpreted as meaning 0–82.5 km.

Page 2, Line10. 'study the composition of Arctic winter' should probably be 'study the composition of the Arctic winter atmosphere' or similar wording.

Page 2, Line 11. 'Sudden Stratospheric Warming' should be all lowercase.

Page 2, Line 20. 'Atmopsheric' should be 'Atmospheric'. This spelling mistake is also made numerous times in the data supplement (https://doi.pangaea.de/10.1594/PANGAEA.861730).

Page 2, Line 31. 'European Centre for Medium Range Weather Forecasts' should be 'European Centre for Medium Range Weather Forecasting'.

Page 5, Lines 27. 'compared with the ACE satellite' should be 'compared with measurements by the ACE-FTS satellite instrument' and the abbreviation ACE (-FTS) defined.

Page 6, Line 10. 'The position relative to the vortex. . .' The position of what?

Page 6, Lines 10–28. In the text the partial columns are stated as over altitudes 40–60 km and 60–80 km whereas in Figure 2 they are 46–66 km and 66–86 km.

Page 7, Line 23. 'X (MLS) and Y (KIMRA)'. The 'X' and 'Y' should probably be replaced by 'abscissa' and 'ordinate' respectively.

Page 9, Line 7. 'movement of higher CO concentrations to lower altitudes' should be 'movement of CO to lower altitudes' or 'CO concentrations (or, more correctly, volume mixing ratio) increased at lower altitudes'.

Figure 1. The subplots should be labelled (a), (b), (c), and (d) and 'n' in the left-hand plot defined. Does the residual plot show measurement minus fit or fit minus
measurement? The axis label '[VMR/VMR]' is confusing as the other axes show units in square brackets (VMR is a quantity); probably the label should be 'Averaging kernel'.

Figure 3. The different curves would be clearer if plotted using different solid / dashed lines with different symbols for each line. What are the units of VMR?

Figure 4. The different curves in (a) would be clearer if they were all plotted using different solid / dashed lines. The differences shown in panel (b) might be clearer if shown as percentages (as is done for temperature in Figure 5(b)).

Figures 5 and 6. The ranges of the horizontal axes in panels (c) and (d) should be adjusted to show the data more clearly.

Figure 7. It would be helpful to mark the occurrence of SSW's. The colorbar label should include 'CO VMR' as well as '[ppmv]'.

---

## Referee Comment (RC2) · Anonymous Referee #2 · 4 Nov 2016

This manuscript presents a mesospheric CO dataset covering the period 2008-2015 obtained by means of a ground-based radiometer, KIMRA, installed at Kiruna. The dataset is very valuable and I recommend its publication on ESSD. There are, however, several points I suggest the authors should address before the manuscript is published.

General comments:

1) This manuscript relies on a previous publication (Hoffmann et al., 2011) for what concerns the error analysis description, but the two datasets (the one published in 2011 and the one described here) have important differences (versions of ARTS and QPACK, temperature and pressure profiles, time interval of data used in the comparison, resulting vertical profiles, and more). A description of the uncertainties characterizing the presented dataset should therefore be added in this manuscript, also to

clarify more precisely which uncertainties and general data characteristics described in Hoffmann et al. apply here as well. This is not clear at the moment.

2) In order to consider this manuscript a validation of this updated and extended dataset, the authors should be more quantitative on the biases with respect to Aura/MLS and show in more details the differences between KIMRA and MLS. It would be important to know and state what the relative difference between the two datasets is, at all (most) altitudes. In the abstract, authors state that the two sets of data have a "high level of agreement", but a possible bias of 10-15% over parts of the vertical coverage does not qualify as "high level of agreement". Figure 4 (b) also shows that in the upper mesosphere (65-85 km) there's a large std dev of the difference (which is not sufficiently stressed in the manuscript) and this weakens the stated agreemeent between the two datasets.

3) The uncertainties and potential biases of the dataset uploaded in the database are not clear. I agree that it is useful to underline how important it is the use of different T and P vertical profiles in the analysis (and hence the presented study on the two different KIMRA set of retrievals), but eventually potential users/readers are interested in knowing how to characterize/treat/use the data made available, e.g., the retrievals associated to the ECMWF reanalysis. As a start, in the various comparisons discussed in the manuscript the authors should always state if the bias is positive or negative, i.e., which dataset is high or low.

4) The altitude range of the dataset that is recommended for scientific use is not clear, as numbers change in the manuscript and differ from the 40-80 km range (with the interval 70-80 km to be used with care) recommended in Hoffmann et al.

5) Hoffmann et al. (2011) explictly stated that the use for that dataset was only for comparison and validation with higher vertical resolution datasets, and therefore the error associated with those profiles did not include, for example, the smoothing error. I personally find this approach correct. Here, however, it is a different matter. Authors

should advice potential users that intend to employ the updated and extended KIMRA dataset as a stand alone product that the uncertainties in the mixing ratio profiles can be much larger due to, for example, the smoothing error. More importantly, I think that in the dataset there should be the information on the estimated total uncertainty, as it is described for example by Hoffmann et al.

Specific comments:

Page 4, Lines 1-2: Is the sinewave removal accounted for in the error estimate and sensitivity of the retrieval? Sinewaves should also be shown in Figure 1 to give the reader an idea of what their amplitude might be compared to the errors listed and the spectral line intensity.

Page 4, Lines 1-2: Same as above.

Page 4, Lines 15: Co-authors should not be cited in personal communication, I think.

Page 4, Lines 27: See general comment #1.

Page 5, Line 10: Looking at the dataset, there are profiles which have a measurement response larger than 0.8 at 93 km altitude. This upper limit for the retrieval is, however, unrealistic. Authors should state explicitly in the manuscript and in the uploaded dataset which is the altitude range where KIMRA profiles are reliable.

Page 5, Line 17: See general comment #1.

Page 5, Line 20: This concept is clear. However, you should also advice potential users that intend to employ your dataset as a stand alone product.

Page 5, Line 31: figure 4e seems to suggest a larger coincidence criteria.

Page 6, Line 15: In the X-title of the plots of figure 2 it says 46-66 km, instead of 40-60. Which is it?

Page 7, Line 4: I am not sure this figure is necessary.

[Figure]

Page 7, Line 10: Please, explain in more details what is this correlation. If you refer exactly to what has been done by Hoffmann et al., please say so.

Page 7, Line 14: In figure 4, the difference between MLS and MLS smooth is not convincing. Kimra averaging kernels are apparently pushing MLS towards larger CO vmr values with respect to its original values. A good set of AKs should just smooth the original dataset, degrade it, not add positive or negative biases. Furthermore, this plot doesn't illustrate what the actual differences between KIMRA and MLS is. X-scale is maybe too large, and it would be useful also to see the relative difference in percentage.

Page 7, Line 20: I understand that this analysis replaces the older one, but maybe a tentative guess as to what is working now that wasn't working before should be attempted. Also, if the analyses are so different, more needs to be said concerning the error analysis of this latest one (see general comment #1).

Page 7, Line 28: This is true only up to 65 km, above that (a large fraction of the profile) Hoffmann et al. show that the predominant error is due to the doppler broedening, e.g., temperature, and in this case doubling the measurement error is definitely not an overestimation, but again an underestimation.

Page 7, Line 29: It's not clear to me why a slope greater than one means greater variation in KIMRA CO concentrations with respect to MLS. Please, explain.

Page 8, Line 2: In order to consider this manuscript a validation of this dataset, the authors should be more quantitative and show in more details the differences between KIMRA and MLS (see general comment #2).

Page 8, Line 11: This sentence is unclear.

Page 8, Line 21: I understand that MSIS profiles from a different time of the day can be different from one another. Yet, looking at figure 5, their difference at 115 km and their low correlation at 100 km are surprising.

Page 9, Line 10: By using your dataset only, how can you distinguish a SSW from a

simple shift of the vortex further away from kiruna?

Page 9, line 13: Remove point after "2015)".

Page 9, Line 19: As commented earlier, what about users of this dataset as a stand alone? Should you add to the dataset also a column with the total uncertainty?

Page 9, Line 23: Is this the official altitude range? What about the considerations in Hoffman et al. that limit the upper altitude to 80 km? Also, in figure 4 (b), are the ïĄĎVMR below 50 km particularly small because the inversions don't have enough sensitivity below 50 km?

References: Straub et al., 2013, is missing.

Figure 2: Units of the X-axis are missing.

[Figure]

---

## Referee Comment (RC3) · Anonymous Referee #3 · 7 Nov 2016

This is an interesting and well written study of CO in the mesosphere above Kiruna. My primary concern with this work is in the CO retrievals. The authors are, apparently for each measurement, fitting 3 sinusoidal baseline waves plus a second order polynomial. This seems to be done as part of a pre-retrieval process. They then go on to do an optimal estimation retrieval, and from that retrieval calculate the "measurement response". But if the baseline waves removed in the pre-retrieval process affect the retrieved mixing ratio profile then the measurements response calculated in the optimal estimation retrieval no long represents the true measurement response. In fact, having removed all of these waves, the authors need to reassure the reader that significant information remains in the retrieval.

In order to address this issue the authors need to do one of several things. 1) They could show (or simply state) that the waves fitted in the pre-retrieval process do not

significantly affect the retrieved vertical profile. 2) They could keep the waves constant for all of the measurements – this would at least give them a measure of the variation of CO if not an absolute measure. 3) They could include fit the baseline waves as part of the optimal estimation retrieval.

The other more serious issue regards Page 8 line 22 - "The slopes of the lines of best fit were calculated without assuming uncertainty in the temperature datasets." I'm not sure what the authors are trying to say here, but any slope fit must always make some assumption about uncertainties. I assume what the authors are doing here is using a generic linear fitting routine, in which case they are implicitly assuming that all of the uncertainty is in the dependent variable. This is almost certainly not appropriate since the comparison is between two types of temperature datasets, neither one of which is perfect. It would be far better to use the same slope fitting routine as was used in the KIMRA v. MLS comparisons and assume that the two temperature datasets have similar precisions.

I do have a few minor comments:

Page 2 line 15 – "offering the advantage of providing measurements during polar night." Advantage relative to whom? There are certainly other satellite instruments that measure during the night. Page 2 line 25 – "a new CO dataset". Perhaps a better phrase here would be "a new CO retrieval" so as not to imply that these are new measurements. Page 3 line 11 – "So the pointing angle changes from one measurement to another, meaning that individual spectra cannot be averaged to reduce the SNR." While one can perhaps understand the author's reluctance to average together spectra taken at different angles, there are certainly many microwave studies which average together spectra taken at different pointing angles. The elevation angle for the retrieval is then based upon an appropriately weighted average of these pointing angles. Spectral averaging periods can vary from as much as a week [Nedoluha et al., JGR, 834-942, 2013] to an hour [Ohyama et al., Earth Planets, and Space, (2016) 68:34] depending upon species and altitude of the retrieval, whereas pointing angle adjustments may occur

over timescales of a minute.

Page 5 line 32 – If the data are filtered to satisfy a DOF > 1 would this result in a high bias since measurements when the CO concentrations are low are more likely to be filtered?

Page 5 line 27 – "In the middle atmosphere the dataset has a positive bias of approximately 20% compared with the ACE satellite, suggested by Livesey et al. (2015) using a validation of the MLS version 2.2 CO data (Pumphrey et al., 2007)." What is not stated here is whether MLS v2.2 CO is biased relative to the MLS v4.2 CO used in this study. Is it?

Page 6 line 32 – "a vertical resolution of more than twice ..." can be a difficult to interpret phrase. Simply adding "...as good as ..." would be helpful, but even better would be some approximate vertical resolution numbers.

Page 9 line 14 – There is a superfluous "." after a reference.

Figure 1, middle: I assume that this is the spectrum after removal of the 3 sine waves and the second order polynomial described on Page 3. If so, this needs to be clearly stated either in the caption or in the text near Page 5 line 7.

———————————————————

---

## Author Comment (AC1) · 24 Dec 2016

**Response to comments from Reviewer #1**

**General comments**

The paper describes a new atmospheric dataset which comprises a time series of carbon monoxide (CO) vertical profiles above Kiruna, northern Sweden. It carefully details the derivation of CO volume mixing ratio profiles from ground based microwave radiometer measurements during 2008 to 2015. The methodology uses well-established atmospheric

10 radiative transfer and retrieval codes and builds on previously reported work by the group on ozone profile measurements using the same radiometer instrument. The effect of a priori temperature on such retrievals can be important, and this is investigated using temperature profiles constructed from various observed and modelled (or reanalysis?) datasets. As well as describing how the profiles were determined and validated the paper indicates how they

15 should be used e.g. applying the supplied averaging kernels to smooth other data-sets when making comparisons. Measurements such as these are important, providing insights into atmospheric dynamics and chemical transport in the polar middle atmosphere and having applications in climate modelling and 'ground-truthing' of satellite data.

Overall the paper is well structured and presented with adequate description of the

20 methodology, discussion of the results, and citing of prior work. However some sections are somewhat challenging to follow, for example the discussion of differences between data-sets where it can be unclear which dataset is positively biased in comparisons with other data. I've highlighted a number of other areas in the text and figures where clarifications and improved presentation are needed. I recommend that the authors address these points before the

25 revised paper is accepted for publication in Earth System Science Data.

**Specific comments**

30 **Title. A date range of 2008–2015 would be better as 'since 2008' is ambiguous. It would be clearer if a more complete description of the measurement location was given in the title, e.g. Kiruna, Sweden including the geographic co ordinates.**

The title has been changed to *"Strato-mesospheric carbon monoxide profiles above Kiruna, Sweden (67.8 N, 20.4 E), since 2008."*

35 Because the measurements are ongoing and the data will likely be updated to include new data, having an "end-date" in the title could deter people from accessing the data if they assume it is limited to up to 2015. Instead, the second line of the abstract has been changed to *"The dataset currently spans the years 2008 to 2015, and measurements are ongoing at Kiruna."* We believe that this immediate clarification, coupled with the publishing date, will make a clear picture of

40 the status of the data.

**Page 1, Line 15. 'CO concentrations' should be 'CO volume mixing ratio (VMR, in ppmv)' here and throughout the manuscript.**

Volume mixing ratio is the only unit of concentration used in the work and so the terms are considered synonymous. The following line of introduction has been modified to explicitly say this.

*"The loss rates in the thermosphere are low and this leads to a strong vertical gradient in CO concentrations (volume mixing ratio (VMR) is the form of gas concentration used throughout this work, and the two terms are considered to be synonymous here)."*

**Page 2, Line 23. 'Leading to a bias of > 5 ppm at 80 km'. Here and throughout the paper it should be made clear what data the biases are between and which of the compared data-sets has the higher values. Also the units of VMR should be ppmv.**

This has been modified to clarify that KIMRA is defined as having a positive bias.

*"The comparisons showed good agreement below 60 km but at higher altitudes the profiles significantly diverged leading to a positive bias in KIMRA CO of > 5 ppmv at 80 km. The shape of the bias in the profile was consistent between satellite datasets and its origin is unclear.."*

Section 3.2 has been modified to clarify that the bias refers to KIMRA minus MLS.

*"The mean profiles show a maximum absolute bias of ~0.65 ppmv at 68 km, and a maximum relative bias of 22% (0.44 ppmv) at 60 km, with bias being defined as KIMRA minus MLS.*

Section 4.1 defines the comparison as *"(ECMWF/MSIS minus/vs. NCEP/SSMIS/MSIS)".*

**Page 2, Line 34–35. 'The timespan of continuing KIMRA measurements will generally surpass the lifetime of any satellite instrument'. This is an important point that could be expanded on, e.g. indicating what timespans / lifetimes are typically achieved by ground based and satellite instruments. What factors allow KIMRA, and other ground based instruments to make measurements that surpass those by individual, or overlapping, satellite missions. Why are long term datasets such as these important?**

The following text has been added to the introduction to elaborate on the point:

*"Many satellite instruments have well exceeded their mission lifetime, the three instruments mentioned in this section being good examples: MLS (2004 - present), ACE-FTS (2003 - present), and MIPAS (2002 - 2012). Ground-based instruments, however, have the potential to produce much longer datasets, albeit at one location, due to the much smaller cost of the ground-based projects and the ability to maintain the instruments. The ground-based radiometer OZORAM has been measuring ozone in the Arctic (79.9˚ N, 11.9˚ E) since 1994 (Palm et al., 2010), and Nedoluha et al. (2016) recently showed twenty years of chlorine monoxide measurements in the Antarctic (77.85˚ S, 166.77˚ E) with the ground-based radiometer ChlOE1. The Fourier transform infrared spectrometer at Kitt Peak (31.9˚ N, 111.6˚ W) (Brault et al., 1978) produces datasets beginning in the 1970's. These are just two examples of a large number of ground-based instruments. Long-term (decades) ground-based datasets are important as they reveal changes in the atmosphere on the time scale of changes in Earth's climate, and the data can be used to validate satellite instruments, fill gaps in time between satellite missions, and to help combine satellite datasets that do not overlap in time."*

**Page 3, Line 13. 'So the pointing angle changes from one measurement to another'. Does this mean that both the elevation and azimuthal angles are changed? If the azimuth is kept fixed while elevation is changed, as I suspect, then the azimuthal angle should be stated. Is the optimum pointing angle checked and, if necessary, changed after each calibration cycle measurement from which a brightness temperature spectrum can be calculated or after a longer time interval? What is the range of pointing angles?**

Section 2.1 has been modified to include the following text that clarifies a range of elevation angles between 5 and 90 degrees, and an azimuth angle of either 0 or 180 degrees. The meaning of a measurement cycle, comprising multiple hot/cold calibrations and producing one time-averaged spectra is also explained.

*"For a given measurement cycle, which includes multiple hot/cold calibrations and produces one time-averaged atmospheric spectrum, KIMRA points to the atmosphere at an elevation angle, between 5˚ and 90˚, that is chosen to provide the best signal-to-noise ratio (SNR) at that time. This angle is governed by the atmospheric conditions and so can change from one measurement cycle to another. Because they are produced using different elevation angles, the individual spectra are not averaged to reduce the SNR. Spectral averaging has been used for similar measurements from other instruments that vary the elevation angle, e.g., over timescales of a week in Nedoluha et al. (2013) and an hour in Ohyama et al. (2016). Rather, each KIMRA spectrum is used to retrieve a CO profile, which may then be used in an average. The azimuth angle of a KIMRA measurement is either 0˚ (directly north) or 180˚ (directly south) and can change from one measurement cycle to another. The azimuth and elevation angle are kept constant during each measurement cycle."*

**Page 3, Lines 12–14. Please clarify whether 'individual spectra' and 'each spectrum' means those calculated from one calibration cycle, or integrated measurements over a longer period.**

This is now clarified in the first sentence of the addition to Section 2.1

*"For a given measurement cycle, which includes multiple hot/cold calibrations and produces one time-averaged atmospheric spectrum, KIMRA points to the atmosphere at an elevation angle, between 5˚ and 90˚, that is chosen to provide the best signal-to-noise ratio (SNR) at that time"*

**Page 3, Lines 20–23. Which version of WACCM is used, for which year is WACCM winter-time output used, and what are the WACCM grid points? If only one winter of WACCM data is used to generate the a priori CO profile is that representative for the years 2008–2015? Is an average profile for winter (presumably December, January, and February) appropriate for all of the months September to May where the CO profile will differ considerably?**

**Page 3, Lines 23–28. The source of the CO a priori data is given but not for the other atmospheric gases included in the forward model. How are the ozone, water vapour, molecular oxygen, nitrogen, and nitric acid a priori profiles constructed?**

Section 2.2 has been modified as follows to include the following information on the WACCM model version, suitability of the a priori, and sources of the profiles for the other gases.

*"The a priori CO information used here is the average of one winter period (September through April) of output from the Whole Atmosphere Community Climate Model, version 4 (WACCM4) (Garcia et al., 2007) provided by Douglas Kinnison at the National Center for Atmospheric Research (NCAR), with a standard deviation of 100% at all altitudes. This combination was found to give enough freedom to the inversion to fit expected changes in CO above Kiruna throughout a given winter period (here September through May) while providing enough regularisation of the retrieved solution so that no oscillations are readily observed in the CO profiles. The WACCM data is on a 132-layer grid between approximately ground and 130 km."…*

*…" A priori profiles of $O_3$, water vapour, and $O_2$ are from the same WACCM run as the CO a priori, and $N_2$ and $HNO_3$ a priori profiles are from the FASCOD subarctic winter scenario (Anderson et al., 1986). A priori profiles of CO, $O_3$, water vapour, $O_2$, and $N_2$ are unchanged from those described in Hoffman et al. (2011 and 2012)."*

**Page 3, Line 32. In this type of microwave measurement baseline errors include, and may be dominated by, those arising from standing waves. However there will be contributions to baseline error from imperfect radiometric calibration, non-linearity, and other instrumental artefacts introduced in the signal down-conversion and spectrometric analysis. While a detailed study of these errors, and their impact on the CO retrieval, is beyond the scope of this paper it would be helpful to indicate them here and later on in the discussion section. The contribution of baseline noise will also depend on the system noise temperature (~1800 K according to Table 1), spectrometer channelbandwidth, and the integration time.**

After recommendation from Reviewer #2, a new section has been added that estimates the error contributions on the profile. Some more information is also added to clarify that the estimations do not cover all instrumental uncertainties but are considered to be the dominant ones.

*"2.4 Error estimates for the retrieved dataset*

*Errors in the retrieved CO profiles arise from uncertainties in instrumental parameters and in parameters that are used as input to the forward model. The relative contributions from these uncertainties are calculated here using the OEM error definitions and by introducing perturbations to the inputs. OEM error definitions are described in detail in Rodgers (2000), among others, and are not repeated here. Figure 2 shows the estimated contributions to the CO profile error budget from the following uncertainties. There are, of course, possible sources of error in any stage of the instrumentation, and the parameters discussed here are considered to represent the dominant uncertainties.*

*The statistical noise, $\Delta T$, on a spectrum is governed by the so-called radiometer equation $\Delta T = T_{sys}/(\Delta \nu \tau)^{1/2}$, with $T_{sys}$ as the system noise temperature of the instrument, $\Delta \nu$ as the channel bandwidth (Table 1), and $\tau$ as the integration time for a measurement of the atmospheric signal. The uncertainty for the temperature profile used in an inversion is the same uncertainty used in Hoffman et al. (2011): 5% below 80 km, 10% above 100 km, and linearly interpolated in between. Uncertainties in the spectroscopic parameters of the CO line are from the HITRAN 2008 catalogue*

*and are: 1% for the line intensity, 2% for the air broadening parameter, and 5% for the temperature dependence of air broadening. Uncertainty in the line position is ignored as the frequency grid can be adjusted to match the line centre in a measurement, and the parameters associated with self-broadening are considered as having negligible impact due to the relatively*
5    *low concentration of the observed gas (Ryan and Walker, 2015). For the blackbody targets used in calibration of the atmospheric measurement, an uncertainty of 2 K is assumed in the temperature of the hot and cold target: a conservative estimate that accounts for fluctuations and drifts in temperature. The uncertainty in the pointing of the instrument to the sky is estimated as 1˚. This is an order of magnitude higher than the precision of the motor that controls pointing,*
10   *to allow for a possible offset in the orientation of KIMRA.*
*The resulting error estimates as well as a sum, in quadrature, of the errors are plotted in Figure 2, and show that the predominant error in the CO profiles comes from the statistical measurement noise on the spectrum. This peaks at 30% of the mean KIMRA CO profile, at 53 km altitude. The error from uncertainties in the temperature profile also shows a significant contribution to the*
15   *total error in the retrievable altitude range above approximately 60 km. The error due to noise on the spectrum is calculated during each inversion and is provided in the supplemental data with the corresponding CO profiles. The error profile in Figure 2 is an average over all measurements. The other plotted errors are calculated about the a priori CO profile and serve as an estimate of the respective error contributions to each measurement. To calculate these errors for individual*
20   *measurements is computationally expensive and so the values plotted in Figure 2 are provided in the supplemental data."*

**Page 4, Line 3. 'in case some changes in the baseline with time became evident.' Did any**
25   **changes in the baseline become evident in processing this dataset?**
The following line has been added to address this point:

*"Sine waves in the baseline are not generally visible by eye on the CO spectra but there were no changes over time found in the determined wave periods. The periods of the fitted waves are 27.5*
30   *MHz, 55 MHz, and 36.3 MHz, with an estimated uncertainty in the amplitudes of 0.5 K, 0.3 K, and 0.5 K, respectively. These calculated sinewave periods are almost identical those found by Hoffman et al. (2011) for the 08/09 and 09/10 winters (27.5 MHz, 55 MHz, and 36.6 MHz), and are similarly large in comparison to the width of the CO spectral line, so that they are uniquely distinguishable from it."*

**Page 4, Line 5. How do the estimated uncertainties (0.5 K, 0.3 K, and 0.5 K) in the standing wave amplitudes compare with the baseline noise of the spectra? Are the standing waves being effectively fitted to the baseline noise level?**
40   Some extra information is added in Section 2.3 to outline the relative standard deviations of the baseline fit amplitude and the residual. The fitted baseline is also now plotted alongside the residual for the example in Figure 1.

*"Figure 1 (b) shows an example spectrum of a CO measurement with KIMRA and the corresponding inversion fit. The inversion fit includes the baseline fit described in Section 2.2 and the example baseline fit is plotted alongside the residual in Figure 1 for comparison. Considering the entire dataset, the standard deviation (averaged across all spectrometer channels) of the amplitude of the fitted baseline is 0.21 K, and the average standard deviation of the residual is 0.34 K. In other words, changes in the retrieved amplitude of the baseline are, on average, lower than the statistical measurement noise on the spectrum."*

**Page 4, Lines 19–20. Why are the KIMRA CO profiles most sensitive to temperature information between approximately 40 km and 90 km? Although the CO line profile will be dominated by thermal Doppler broadening in the upper mesosphere, at lower altitudes pressure broadening will dominate. This could also be discussed in the context of the averaging kernels of the CO retrieval and estimated altitude resolution (Figure 1 and Page 5, Lines 7–20).**
This was badly worded. The sentence was supposed to indicate that the sensitivity of the KIMRA CO retrieval to the temperature input is largest within the retrievable altitude range. This is because the measurement response is largest within this range. The wording has been changed to clarify this.

*"The inversions utilising SSMIS data are considered as those using the most suitable available data for the CO inversion, as the sensitivity of KIMRA CO profiles to atmospheric temperature information is strongest within the retrievable altitude range (on average between 48 km and 84 km), and the resulting CO dataset is considered as a reference point for inversion setups using alternate input temperature information."*

Section 2.3 now includes information relating to the averaging kernels and Doppler broadening in the atmosphere.
*"The centres of the averaging kernels, when represented in VMR, are shifted down in altitude compared to a representation using relative concentrations. Hoffmann et al. (2011) provide a detailed discussion of the representation of averaging kernels for ground-based CO measurements, and the major points are worth repeating here. While here the lower limits of the retrievable altitude ranges are set by the SNR of the measurement, the upper limit of the measurements is set by the transition from a pressure broadening regime to a Doppler broadening regime (not considering spectrometer channel resolution). The result of this is that, above approximately 70 km in the VMR representation, the altitude locations of the centres of the averaging kernels do not increase anymore with the increase in respective retrieval altitude. And while the retrieved CO values above 70 km do contain information from the atmosphere corresponding to the retrieval altitude, the VMR representation of the data above this altitude should be considered with care."*

**Page 7, Line 6–7. MLS values at 82 km and 84 km are stated as often unusable for scientific work. Are MLS data with the warning flag at these altitudes included in the analysis and, if so, how might that affect the results?**
Text has been added to this section to clarify:

*"MLS values at 82 km and 84 km are often defined as unusable for scientific work due to a too-high a priori contribution (Livesey et al., 2015) and the precision is given as negative in this case. The CO values are still considered here as useful for comparative purposes, but quantities derived using the precision (e.g., the slope of a line) are not meaningful at these altitudes."*

**Page 9, Line 4. Do you really mean the data from 2015/16 are presently unavailable? This suggests the data might become available in future, whereas my understanding of 'failure of the KIMRA cooling system' is that measurements were not collected during this period.**

10  While the cooling system failed, it is possible to run KIMRA with the mixer at room temperature. But with a much higher noise temperature, the measurements of CO are not really feasible. As such, the line in the manuscript is changed to just say "unavailable".

15  **Page 9, Lines 10–13. What is happening to the CO during and after SSWs? What is the difference between major and minor SSWs and is the observed CO behave as would be expected for these events. What might be causing large decreases in CO abundance above _65 km during December 2013 (12/13 on Figure 7), that is not assigned to a SSW?**

Information is now provided on major and minor SSWs, and an influx of lower latitude air during
20  the destabilization of the polar vortex is given as the reason for the decrease in CO concentrations. A note to refer to Section 1 has been included after *"…and then the subsequent increases as the vortex recovered …"* to refer to the high CO concentrations associated with the polar vortex.

There are many artifacts of the data that are not interpreted in this work because it is difficult to
25  do so without external information, and so only some general comments are made with the help of the referenced material. This is referred to in the beginning of the second paragraph and we would prefer not to offer speculation. A full interpretation of the results falls outside of the scope of this work but he text has been edited to clarify that there are many visible fluctuations in the data that are not interpreted here.

*"Signatures of "major" SSWs (during which the 10 hPa circulation becomes easterly at 60˚ N), beginning 24ᵗʰ January, 26ᵗʰ January, and 6ᵗʰ January, 2009, 2010, and 2013, respectively, can be seen by the quickly (order of days) decreasing CO concentrations around these dates, and then the subsequent increases as the vortex recovered (see Section 1) (Manney et al., 2009, 2015). The*
35  *effects of a "minor" SSW (during which the 10 hPa circulation remains westerly at 60˚ N), in early January 2015 (Manney et al., 2015) can also be seen. Decreases in CO concentrations during SSWs are mainly due to the influx of lower latitude air as the polar vortex destabilises. There are other visible fluctuations in the presented KIMRA CO data over various timescales which, while not interpreted here, can be used in the characterisation of winter-time dynamics above Kiruna."*

40

**Technical comments**
**Page 1, Lines 17–18. 'This dataset is compared. . .' It should be made clear which dataset is being referred to here.**

The line has been changed to say "*This KIMRA CO dataset…*".

**Page 1, Line 24. 'at all altitudes below 82.5 km'. The exact altitude range should be given as this could be misinterpreted as meaning 0–82.5 km.**
This has been changed to *"A comparison of the two overlapping KIMRA CO datasets shows a bias of < 5% and a correlation > 0.98 between the lower retrievable altitude limit and 82.5 km."*

**Page 2, Line10. 'study the composition of Arctic winter' should probably be 'study the composition of the Arctic winter atmosphere' or similar wording.**
This has been fixed: *"…study the composition of Arctic winter atmosphere in 2001/2002…"*

**Page 2, Line 11. 'Sudden Stratospheric Warming' should be all lowercase.**
This has been fixed.

**Page 2, Line 20. 'Atmopsheric' should be 'Atmospheric'. This spelling mistake is also made numerous times in the data supplement (https://doi.pangaea.de/10.1594/PANGAEA.861730).**
One would think I would be able to spell atmosphere at this point…
This has been fixed in the manuscript and will be changed in the data supplement when resubmitting the data with a more complete error estimate.

**Page 2, Line 31. 'European Centre for Medium Range Weather Forecasts' should be 'European Centre for Medium Range Weather Forecasting'.**
This has been fixed.

**Page 5, Lines 27. 'compared with the ACE satellite' should be 'compared with measurements by the ACE-FTS satellite instrument' and the abbreviation ACE (-FTS) defined.**
This has been fixed. ACE-FTS is defined in section 1.

**Page 6, Line 10. 'The position relative to the vortex. . .' The position of what?**
This has been edited to *"The location of a measurement, its position relative to the vortex, and the distance between measurements were calculated at 50 km altitude."*

**Page 6, Lines 10–28. In the text the partial columns are stated as over altitudes 40–60 km and 60–80 km whereas in Figure 2 they are 46–66 km and 66–86 km.**
The mistake was in the text. The text has been changed to indicate the same ranges as Figure 2.

**Page 7, Line 23. 'X (MLS) and Y (KIMRA)'. The 'X' and 'Y' should probably be replaced by 'abscissa' and 'ordinate' respectively.**
This has been done.

**Page 9, Line 7. 'movement of higher CO concentrations to lower altitudes' should be 'movement of CO to lower altitudes' or 'CO concentrations (or, more correctly, volume mixing ratio) increased at lower altitudes'.**
10   This has been changed.

**Figure 1. The subplots should be labelled (a), (b), (c), and (d) and 'n' in the lefthand plot defined. Does the residual plot show measurement minus fit or fit minus measurement? The axis label**
15   **'[VMR/VMR]' is confusing as the other axes show units in square brackets (VMR is a quantity); probably the label should be 'Averaging kernel'.**
These changes have been made to the figure. The residual is the measurement minus the inversion fit and this is now stated in the caption.

**Figure 3. The different curves would be clearer if plotted using different solid / dashed lines with different symbols for each line. What are the units of VMR?**
Different line styles with different symbols have been used and the x-scale has been shortened. It's hard to see differences in the profiles below 60 km but the aim of the plot is to show how the
25   MLS profile has been altered.The caption now reads:
*"Figure 3. The mean of the coincident MLS CO profiles and the a priori CO profile used for the KIMRA inversion (Section 2.2). MLS orig. shows the mean of the supplied MLS profiles, which use a constant CO concentration above 0.001 hPa (~ 98 km). MLS extrap. shows the mean of the profiles that have been linearly extrapolated in pressure space above 0.001 hPa to provide more*
30   *physical CO concentrations at these altitudes."*

**Figure 4. The different curves in (a) would be clearer if they were all plotted using different solid / dashed lines. The differences shown in panel (b) might be clearer if shown as**
35   **percentages (as is done for temperature in Figure 5(b)).**
These changes have been made to Figure 4.

**Figures 5 and 6. The ranges of the horizontal axes in panels (c) and (d) should be adjusted to**
40   **show the data more clearly.**
The axes have been adjusted to better show the range of the data.

**Figure 7. It would be helpful to mark the occurrence of SSW's. The colorbar label should include 'CO VMR' as well as '[ppmv]'.**

The label of the colourbar has been changed.

While it is agreed that it would make it easier to pinpoint the dates mentioned in Section 4.2 on Figure 7 (now Figure 8), it is preferred to not mark the dates of SSW's, or any other phenomena. While the data can be used to characterize changes in CO above Kiruna in the context if a SSW, it is not the only/primary use of the data and the marks in the plot would tend to overemphasize SSWs.

**Strato-mesospheric carbon monoxide profiles above Kiruna, Sweden (67.8° N, 20.4° E), since 2008**

[revised manuscript text omitted]

This paper presents a CO dataset from KIMRA, using an extended set of measurements, beginning in 2008, and retrieved using a new inversion setup. The layout of the paper is as follows: Section 2 provides details on the KIMRA observation system: the measurements and the inversion technique. Section 2 also offers an estimation of the errors in CO profiles, and a word on interpretation of the data. Section 3, to establish validity of the observation system, presents a comparison of the KIMRA data with MLS data, using temperature information from the Special Sensor Microwave Imager/Sounder (SSMIS) (Kunkee et al.,2008; Swadely et al., 2008) aboard the US Air Force's Defense Meteorological Satellite Program F-18 satellite as input for the KIMRA CO inversion. Section 4 investigates changes in the retrieved KIMRA CO data when using temperature information from the European Centre for Medium-Range Weather Forecasting (ECMWF), and presents this extended dataset, currently spanning 2008-2015. Section 5 concerns the availability and use of the data, and Section 6 offers some concluding remarks.

Satellite data is exchanged for ECMWF analysis in the KIMRA inversion in order to have a consistent temperature input for the entire KIMRA dataset; The timespan of continuing KIMRA measurements will generally surpass the

lifetime of any satellite instrument. Many satellite instruments have well exceeded their mission lifetime, the three instruments mentioned in this section being good examples: MLS (2004 - present), ACE-FTS (2003 - present), and MIPAS (2002 - 2012). Ground-based instruments, however, have the potential to produce much longer datasets, albeit at one location, due to the much smaller cost of the ground-based projects and the ability to maintain the instruments.

5      The ground-based radiometer OZORAM has been measuring ozone in the Arctic (79.9° N, 11.9° E) since 1994 (Palm et al., 2010), and Nedoluha et al. (2016) recently showed twenty years of chlorine monoxide measurements in the Antarctic (77.85° S, 166.77° E) with the ground-based radiometer ChlOE1. The Fourier transform infrared spectrometer at Kitt Peak (31.9° N, 111.6° W) (Brault et al., 1978) produces datasets beginning in the 1970's. These are just two examples of a large number of ground-based instruments. Long-term (decades) ground-based datasets are

10     important as they reveal changes in the atmosphere on the time scale of changes in Earth's climate, and the data can be used to validate satellite instruments, fill gaps in time between satellite missions, and to help combine satellite datasets that do not overlap in time.

**2 Instrument and dataset**

**2.1 KIMRA**

15     KIMRA is housed at the Swedish Institute for Space Physics (IRF), Kiruna, and was partly designed by the Institute for Meteorology and Climate Research (IMK) at the Karlsruhe Institute of Technology (KIT) (Raffalski et al., 2002). KIMRA utilises the frequency range of 195 – 233 GHz and has been measuring, among others, atmospheric spectra that correspond to the J = 2 → 1 rotational transition (230.54 GHz) of CO. KIMRA has operated in Kiruna since 2002 and has been making measurements of CO emissions since 2008. The more general aspects of the instrument are given

20     in Table 1 and the more specific details can be found in Raffalski et al. (2002) and Hoffmann et al. (2011). The spectrometer used for CO measurements is a Fast Fourier Transform Spectrometer (FFTS) made by Omnisys Instruments, with 1024 channels and used with a bandwidth of 110 MHz to give a resolution of ~107 kHz per channel. For a given measurement cycle, which includes multiple hot/cold calibrations and produces one time-averaged atmospheric spectrum, KIMRA points to the atmosphere at an elevation angle, between 5° and 90°, that is chosen to

25     provide the best signal-to-noise ratio (SNR) at that time. This angle is governed by the atmospheric conditions and so can change from one measurement cycle to another. Because they are produced using different elevation angles, the individual spectra are not averaged to reduce the SNR. Spectral averaging has been used for similar measurements from other instruments that vary the elevation angle, e.g., over timescales of a week in Nedoluha et al. (2013) and an hour in Ohyama et al. (2016). Rather, each KIMRA spectrum is used to retrieve a CO profile, which may then be used

30     in an average. The azimuth angle of a KIMRA measurement is either 0° (directly north) or 180° (directly south) and can change from one measurement cycle to another. The azimuth and elevation angle are kept constant during each measurement cycle. Figure 1 (a) shows the distribution of the duration times of each measurement.

**2.2 Inversion setup**

CO profiles are retrieved from the spectra using an optimal estimation method (OEM) inversion (Rodgers, 2000). This is a Bayesian statistical approach that constrains the retrieved CO profile according to some a priori atmospheric information. The inversion was carried out using the Qpack 2 (Eriksson et al., 2005) package, which employs the Atmospheric Radiative Transfer Simulator: ARTS 2 (Eriksson et al., 2011) to model radiative transfer through the atmosphere, i.e., the forward model. All of the following information that is input into the inversion is done so using Qpack 2. The a priori CO information used here is the average of one winter period (September through April) of output from the Whole Atmosphere Community Climate Model, version 4 (WACCM4) (Garcia et al., 2007) provided by Douglas Kinnison at the National Center for Atmospheric Research (NCAR), with a standard deviation of 100% at all altitudes. This combination was found to give enough freedom to the inversion to fit expected changes in CO above Kiruna throughout a given winter period (here September through May) while providing enough regularisation of the retrieved solution so that no oscillations are readily observed in the CO profiles. The WACCM data is on a 132-layer grid between approximately ground and 130 km. Ozone ($O_3$) is also retrieved simultaneously with CO, as there is an $O_3$ spectral line located at 231.28 GHz, and attenuation of the CO spectral line due to water vapour is accounted for by including the water vapour continuum described by Rosenkranz (1998) in the forward model and inversion. The spectroscopic information used is from the HITRAN 2008 catalogue (Rothmann et al., 2009). Continua of molecular oxygen ($O_2$) and nitrogen ($N_2$) (Rosenkranz, 1993), and nitric acid ($HNO_3$) lines are also included in the inversion but are not retrieved and are considered model parameters. A priori profiles of $O_3$, water vapour, and $O_2$ are from the same WACCM run as the CO a priori, and $N_2$ and $HNO_3$ a priori profiles are from the FASCOD subarctic winter scenario (Anderson et al., 1986). A priori profiles of CO, $O_3$, water vapour, $O_2$, and $N_2$ are unchanged from those described in Hoffman et al. (2011 and 2012).

Measurement noise (statistical noise on the spectrum) was estimated by fitting a second order polynomial to a wing of the spectrum and calculating the standard deviation of the fit. As there is no windowing applied in the operation of the FFTS, the spectrometer channels are specified in the inversion as having a sinc-squared response function (Harris, 1978). Three sine wave functions are fitted to the baseline of each spectrum during an inversion to account for errors in the baseline, which are most often produced by standing waves in the instrument. A fitting of functions to the baseline of the measurement (baseline fit) can be included in the optimal estimation performed by Qpack 2 and forms part of the general fit to the measurement (inversion fit). For sine waves, the period and estimated amplitude uncertainty are provided as input, and the amplitude and phase of the waves are retrieved. The periods of the sine waves in the KIMRA spectra were found by first inverting all of the measurements without a fit to the baseline and then evaluating a periodogram of the residuals. This procedure was also applied to subsets of the data in case some changes in the estimated periods with time became evident. Sine waves in the baseline are not generally visible by eye on the CO spectra but there were no changes over time found in the determined wave periods. The periods of the fitted waves are 27.5 MHz, 55 MHz, and 36.3 MHz, with an estimated uncertainty in the amplitudes of 0.5 K, 0.3 K, and 0.5 K, respectively. These calculated sinewave periods are almost identical to those found by Hoffman et al. (2011) for the 08/09 and 09/10 winters (27.5 MHz, 55 MHz, and 36.6 MHz), and are similarly large in comparison to the width of the CO spectral line, so that they are uniquely distinguishable from it. A second order polynomial is also

[revised manuscript text omitted]

Figure 1 (b) shows an example spectrum of a CO measurement with KIMRA and the corresponding inversion fit. The inversion fit includes the baseline fit described in Section 2.2 and the example baseline fit is plotted alongside the residual in Figure 1 for comparison. Considering the entire dataset, the standard deviation (averaged across all spectrometer channels) of the amplitude of the fitted baseline is 0.21 K, and the average standard deviation of the residual is 0.34 K. In other words, changes in the retrieved amplitude of the baseline are, on average, lower than the statistical measurement noise on the spectrum.

The mean of the averaging kernels for the CO dataset are also shown in Figure 1 (c) along with the measurement response (d) (sum of the rows of the averaging kernel matrix) and the altitude resolution of the profiles (full-width-at-half-maximum of the averaging kernels). Altitudes with a measurement response greater than 0.8 are considered to represent the range of useful profile information: the retrievable altitude range. This choice is somewhat arbitrary, with 0.8 being a regularly used value (e.g., Forkman et al., 2012; Hoffmann et al., 2011; Straub et al., 2013). The CO profiles here have an average retrievable altitude range of 48–84 km and an average vertical resolution of between 15 and 18.5 km depending on the altitude. These compare to an average altitude range of 40-80 km and average resolution of 16-22 km for the CO profiles shown in Hoffmann et al. (2011) for the winters of 08/09 and 09/10. DOFS for the current dataset have a mean of 2.0 and a standard deviation of 0.6. The minimum and maximum of the lower and upper retrieval limit for the current dataset is 35 km and 99 km, respectively, but the upper limit of any profile (defined using measurement response) must be considered with the following caveat.

The centres of the averaging kernels, when represented in VMR, are shifted down in altitude compared to a representation using relative concentrations. Hoffmann et al. (2011) provide a detailed discussion of the representation of averaging kernels for ground-based CO measurements, and the major points are worth repeating here. While here the lower limits of the retrievable altitude ranges are set by the SNR of the measurement, the upper limit of the measurements is set by the transition from a pressure broadening regime to a Doppler broadening regime (not considering spectrometer channel resolution). The result of this is that, above approximately 70 km in the VMR representation, the altitude locations of the centres of the averaging kernels do not increase anymore with the increase in respective retrieval altitude. And while the retrieved CO values above 70 km do contain information from the atmosphere corresponding to the retrieval altitude, the VMR representation of the data above this altitude should be considered with care.

**2.4 Error estimates for the retrieved dataset**

Errors in the retrieved CO profiles arise from uncertainties in instrumental parameters and in parameters that are used as input to the forward model. The relative contributions from these uncertainties are calculated here using the OEM error definitions and by introducing perturbations to the inputs. OEM error definitions are described in detail in Rodgers (2000), among others, and are not repeated here. Figure 2 shows the estimated contributions to the CO profile error budget from the following uncertainties. There are, of course, possible sources of error in any stage of the instrumentation, and the parameters discussed here are considered to represent the dominant uncertainties.

The statistical noise, $\Delta T$, on a spectrum is governed by the so-called radiometer equation $\Delta T = T_{sys}/\sqrt{\nu\tau}$, with $T_{sys}$ as the system noise temperature of the instrument, $\Delta\nu$ as the channel bandwidth (Table 1), and $\tau$ as the integration time for a measurement of the atmospheric signal. The uncertainty for the temperature profile used in an inversion is the same uncertainty used in Hoffman et al. (2011): 5% below 80 km, 10% above 100 km, and linearly interpolated in between. Uncertainties in the spectroscopic parameters of the CO line are from the HITRAN 2008 catalogue and are: 1% for the line intensity, 2% for the air broadening parameter, and 5% for the temperature dependence of air broadening. Uncertainty in the line position is ignored as the frequency grid can be adjusted to match the line centre in a measurement, and the parameters associated with self-broadening are considered as having negligible impact due to the relatively low concentration of the observed gas (Ryan and Walker, 2015). For the blackbody targets used in calibration of the atmospheric measurement, an uncertainty of 2 K is assumed in the temperature of the hot and cold target: a conservative estimate that accounts for fluctuations and drifts in temperature. The uncertainty in the pointing of the instrument to the sky is estimated as 1°. This is an order of magnitude higher than the precision of the motor that controls pointing, to allow for a possible offset in the orientation of KIMRA.

The resulting error estimates as well as a sum, in quadrature, of the errors are plotted in Figure 2, and show that the predominant error in the CO profiles comes from the statistical measurement noise on the spectrum. This peaks at 30% of the mean KIMRA CO profile, at 53 km altitude. The error from uncertainties in the temperature profile also shows a significant contribution to the total error in the retrievable altitude range above approximately 60 km. The error due to noise on the spectrum is calculated during each inversion and is provided in the supplemental data with the corresponding CO profiles. The error profile in Figure 2 is an average over all measurements. The other plotted errors are calculated about the a priori CO profile and serve as an estimate of the respective error contributions to each measurement. To calculate these errors for individual measurements is computationally expensive and so the values plotted in Figure 2 are provided in the supplemental data.

**2.5 Smoothing error and interpretation of the KIMRA profiles**

Smoothing error in the profiles arises from the limited vertical resolution of the retrieved profile and can be calculated with the OEM using the averaging kernels and a priori information for a profile. Smoothing error can be large for ground-based profiling instruments due the small altitude spacing between retrieval grid points, chosen here for numerical stability in the inversion (Eriksson, 1999), compared to the actual vertical resolution of the retrieved profiles (see Figure 1 (c) and (d)). Smoothing error should be assessed when one wishes to use or interpret the CO profiles without consideration of the accompanying averaging kernels. As this is not a recommended use of the data, the smoothing error is not assessed here. If using KIMRA profiles to say something of the absolute value of CO at a given retrieval altitude, one must be aware that a CO value at a retrieval grid point contains information from a range of altitudes with a sensitivity governed by the shape of the corresponding averaging kernel.

Smoothing error can be accounted for when comparing KIMRA CO profiles to those of an instrument/model that has a significantly different vertical resolution by using the KIMRA averaging kernel matrices and a priori to smooth (Rodgers and Connor, 2013) the other profiles so that they have a similar vertical resolution. KIMRA profiles can be used to observe changes in CO concentrations over time, provided there is not a significant difference in the averaging

kernels (and thus measurement response) of the profiles over that time. In particular, care should be taken with data near the edges of the retrievable altitude range (where the measurement response is decreasing towards 0.8), as the measurement response in this region can change quickly when there are sharp changes in atmospheric CO.

**3 Comparison with MLS**

This section presents a comparison of CO profiles from KIMRA and MLS that are colocated in space and time. A description of the MLS instrument is given in Walters et al. (2006). Version 4.2 of the CO data (Schwartz et al., 2015) is used here, a description of which can be found in Livesey et al. (2015). Version 4.2 CO data covers the pressure range of 215-0.0046 hPa. The precision of the CO profile reaches a maximum (largest) value of 1.1 ppmv at the highest retrieval layer (0.0046 hPa). In the middle atmosphere the dataset has a positive bias of approximately 20% compared with the ACE-FTS satellite instrument. This estimate is given by Livesey et al. (2015) using a validation of the MLS version 2.2 CO data (Pumphrey et al., 2007), which showed a positive bias of approximately 30%: Later versions than 2.2 show slight lowering of the MLS values, bringing them closer to the ACE-FTS data.

**3.1 Colocation of KIMRA and MLS measurements**

For a given KIMRA measurement, a coincident MLS measurement was defined as follows. MLS measurements that were made within 4 hours, ±2° latitude and ±10° longitude of the KIMRA measurement, and lie in the same position relative to the vortex edge as the KIMRA measurement (inside, outside, or within the edge of the polar vortex) were identified. The MLS measurement from this group that was closest, along a great circle, to the KIMRA measurement was chosen as coincident. A given KIMRA/MLS measurement could only be considered coincident with one MLS/KIMRA measurement. The location of a measurement with respect to the polar vortex was determined using scaled potential vorticity (sPV) values from NASA's Global Modeling and Assimilation Office's (GMAO's) MERRA (Modern Era Retrospective analysis for Research and Applications) (Rienecker et al., 2011). The sPV values for KIMRA were calculated geometrically along the instrument's line of sight. sPV values of 1.6 and 1.2 $x10^{-4}$ $s^{-1}$ have been used extensively in previous works (e.g. Manney et al., 1994, 2007, 2011; Jin et al., 2006) to define the respective inner and outer edge of the vortex, and the same values are used here.

[revised manuscript text omitted]

The mean profiles show a maximum absolute bias of ~0.65 ppmv at 68 km, and a maximum relative bias of 22% (0.44 ppmv) at 60 km, with bias being defined as KIMRA minus MLS. The standard deviation of the differences in

the profiles peaks at 21% at 60 km. These standard deviation values are similar in magnitude to the estimated uncertainties in the KIMRA CO profiles (Figure 2). The standard error on the bias is not shown in Figure 5 (b) and (c) because it is very small due to the number of coincidences, but rather the standard deviation of the differences is shown as the whiskers on the bias. The combination defines a one-sigma space in which a KIMRA profile lies with respect to an MLS profile. The correlation of the profiles is high at all altitudes, only dropping below 0.90 above 82 km. The correlation between KIMRA and unsmoothed MLS profiles is also plotted, with values between 0.81 and 0.90. Previous CO retrievals for winter 08/09 and 09/10 (Hoffmann et al., 2011) showed a bias with respect to MLS, increasing with altitude, with a value of > 5 ppm at 80 km. The structure of this bias was also shown in comparisons of KIMRA with ACE-FTS and MIPAS, and it appears that this attribute is not present in the retrievals presented here. The slope of a line of best fit to KIMRA vs. MLS measurements was calculated individually for each altitude layer as follows: for a given retrieval grid point the slope and intercept (or regression coefficients) for a line of best fit to the KIMRA and MLS values was calculated, accounting for errors in the abscissa (MLS) and ordinate (KIMRA) values according to York et al. (2004). Two cases of KIMRA CO error estimates were used when calculating a line of best fit: the first being the measurement error in the profile (the error due to statistical noise on the spectrum (Section 2.4)), and the second being twice the measurement error. The former is an underestimation of the true error on the profile as there are more error sources than measurement error alone, and the latter is likely an overestimation of the true error as the measurement error is a predominant source of error in the profile (Figure 2). The results (Figure 5 (e)) show that the slope is always greater than the ideal value of 1.0 for both KIMRA error estimates, meaning that KIMRA shows a greater range of CO concentrations at all altitudes and the variation is not explained by the estimated random errors in the profile. The reason for the difference could be due to errors, e.g. spectroscopic information, calibration, baseline wave signatures, that can have a contribution that is neither truly systematic nor random. The difference in calculated slopes for the two KIMRA error estimates is insignificant within the standard error, likely due to the large natural variation in CO concentrations. Despite the > 1.0 slope values, KIMRA and MLS are considered to show agreement, according to the level of difference and correlation between profiles.

**4 Extension of the KIMRA dataset**

After establishing a reliable inversion scheme through comparison with MLS, the KIMRA dataset is extended in time by substituting ECMWF Operational Analyses model output for the SSMIS temperature data in the inversion (see Section 2.2).

**4.1 The effect of temperature input on KIMRA CO profiles**

ECMWF temperatures are available four times per day, in six-hourly intervals beginning at midnight. The same filtering procedure for the retrieved data is employed as outlined in section 2.3. To evaluate the effect of using a different temperature dataset as input to the inversion, the two KIMRA datasets are compared where they overlap between January 2011 to May 2014 and the results are shown in Figures 6 and 7.

A comparison of the two sets of temperature profiles (ECMWF/MSIS minus/vs. NCEP/SSMIS/MSIS) is shown in Figure 6. (a) and (b) show the mean and standard deviation of the differences between temperature profiles in absolute

and relative units, (c) shows the correlation at each altitude, and (d) shows the slope of the lines of best fit at each altitude. The same is shown for the two respective sets of CO profiles in Figure 7. No smoothing with averaging kernels is applied to the data.

The bias for the temperature profiles is very low below 50 km, showing good agreement between ECMWF and NCEP output, as well as the lower altitude SSMIS data, and then moves to a minimum of ~ -4% at 68 km. The maximum in the bias is about 4% at 118 km. The correlation is high below 50 km but has minima of < 0.50 at ~ 70 km and ~ 0.80 at 100 km. It should be noted that while MSIS is used in both temperature datasets, the time of the MSIS output is governed by the times for the ECMWF output and the SSMIS measurements, and so the high-altitude (> 0.01 hPa) temperature values are not necessarily equal for the two inversion setups. The slopes of the lines of best fit were calculated using the same temperature error estimate, as described in Section 2.4, for each dataset. The slope is within 11% of 1.0 below 50 km altitude, above which it decreases to around 0.65 at 56 km, before increasing to 1.4 at 66 km, and then varies about 1 with another peak of 1.3 at 102 km.

There is a general positive bias in the CO profiles that use ECMWF/MSIS, seen in (a) and (b) of Figure 7. The bias is small, reaching a maximum of ~ 5% in the range of 68-78 km. The correlation of the profiles is very high, greater than 0.98 at all altitudes below 82.5 km. The slopes of the lines of best fit were calculated with the same error estimates described in Section 3.2. A value of 1.0 lies in the range of standard error of the slope below 56 km and above 80 km, and between these altitudes reaches a maximum of 1.06. As the only difference in the inversion setups is the temperature input, it follows that any inequalities of the respective KIMRA CO profiles are ultimately due to this difference. Overall, the CO profiles using the differing temperature inputs shown here agree very well.

**4.2 KIMRA CO dataset from 2008 to 2015**

Figure 8 shows the KIMRA CO dataset between December 2008 and May 2015. Daily averaged CO concentrations between 46 and 86 km are shown. Data gaps in this time range are due to non-operation of the instrument or a lack of CO spectral line measurements. Data from winter 2015/2016 are unavailable due to a failure of the KIMRA cooling system.

While it is impossible to fully characterise the concentrations shown without inclusion of other instrument data and/or model output, some observations are made here. The beginning of each winter (from about September through November) shows a movement of CO to lower altitudes, which can in general be expected as predominantly due to vertical advection (Allen et al., 1999; Minschwaner et al.,2010; Solomon et al., 1985). The high CO concentrations remain for most of winter before decreasing again from March onwards, generally due to loss of CO because of increased ·OH and movement of low-CO air from lower latitudes as the final warming of the pole occurs. Signatures of "major" SSWs (during which the 10 hPa zonal circulation becomes easterly at 60˚ N), beginning 24[th] January, 26[th] January, and 6[th] January, 2009, 2010, and 2013, respectively, can be seen by the quickly (order of days) decreasing CO concentrations around these dates, and then the subsequent increases as the vortex recovered (see Section 1) (Manney et al., 2009, 2015). The effects of a "minor" SSW (during which the 10 hPa zonal circulation remains westerly at 60˚ N), in early January 2015 (Manney et al., 2015) can also be seen. Decreases in CO concentrations during SSWs are mainly due to the influx of lower latitude air as the polar vortex destabilises. There are other visible

fluctuations in the presented KIMRA CO data over various timescales which, while not interpreted here, can be used in the characterisation of winter-time dynamics above Kiruna.

**5. Data availability**

The current KIMRA CO dataset (between December 2008 and May 2015), can be accessed publically through PANGAEA Data Publisher for Earth and Environmental Science at: https://doi.pangaea.de/10.1594/PANGAEA.861730. Metadata is also provided including the averaging kernel matrices for each measurement. It is recommended to use the averaging kernels specific to each CO profile when using the data in a comparison with model output or a dataset with significantly different altitude resolution.

**6. Conclusion**

The aim of this work was to create a self-consistent dataset of strato-mesospheric CO profiles above Kiruna, using measurements from the ground-based microwave radiometer, KIMRA, and an optimal estimation technique. The resulting profiles cover an average altitude range of 48-84 km. The retrievable altitude limits vary with the SNR of each measurement, and CO VMRs above 70 km should be treated with care due an offset in centres of the corresponding averaging kernels above this altitude. As a test of the validity of the KIMRA observing system and to create a reference dataset, CO was first retrieved using atmospheric temperature information from the SSMIS satellite instrument, available within 2011-2014, and compared to CO data from MLS. The instruments show agreement, with KIMRA showing a maximum bias of ~ 0.65 ppmv at 68 km (corresponding to 14.7% of the mean CO value at 68 km), and a maximum relative bias of 22% (0.44 ppmv) at 60 km. Correlations with MLS are greater than 0.90 at most altitudes. KIMRA shows a larger range of atmospheric CO concentrations, compared to MLS, that is not explained by the estimates of random errors in the data, and may be due to some combination of random and systematic uncertainty. Some differences in the KIMRA and MLS data can be expected due to imperfect colocation of the measurements and because MLS has a horizontal resolution of 200-300 km in the mesosphere and stratosphere: an order of magnitude wider than the beamwidth of KIMRA at these altitudes. The KIMRA dataset is extended in time (2008 - 2015) by substituting ECMWF Operational Analyses temperature output in place of the SSMIS data. The extended KIMRA dataset shows a difference (bias) of less than 5% compared to the reference dataset (the one using SSMIS data), and correlations between the two are greater than 0.98 at most altitudes. There is a larger range ($\leq 6\%$) of concentrations seen by the extended dataset, compared to the reference dataset over the same time period. The extended dataset currently spans the time between December 2008 to May 2015, with data gaps. Measurements are ongoing at IRF Kiruna.

**Author contribution**

M. Palm designed and took an active role in the project. N. Ryan developed the inversion setups for KIMRA and performed the comparisons. R. Larsson provided SSMIS temperature data. G. Manney and L. Millán provided sPV

information for KIMRA measurements. U. Raffalski maintains KIMRA and provided the KIMRA measurement data. J. Notholt provided valuable feedback on the project. N. Ryan prepared the manuscript with contributions from co-authors.

**Acknowledgements**

5    This work has been funded by the German Federal Ministry of Education and Research (BMBF) through the research project: Role Of the Middle atmosphere in Climate (ROMIC), sub-project: ROMICCO, project number: 01LG1214A. We would like to express our gratitude to the MLS teams for making their CO and sPV products available. We would also like to thank the ECMWF and NCEP teams for making their products available, as well as the Qpack and ARTS communities for making their software available. Work at the Jet Propulsion Laboratory, California Institute of
10   Technology was done under contract with the National Aeronautics and Space Administration.

[revised manuscript text omitted]

**Figure 1: (a)** A histogram of the KIMRA CO measurement durations from January 2011 to May 2014 with n as the number of measurements (see Section 2.2). **(b)** Upper: An example measurement from November 5[th] 2012 with the corresponding inversion fit (which includes the baseline fit, see Section 2.2). Lower: the residual (measurement minus inversion fit), and the baseline fit for comparison. **(c)** The mean averaging kernels for all CO measurements, with the measurement response divided by 4 shown in black. The dashed and dotted black lines indicate a measurement response of 0.8 and 1.0, respectively. **(d)** The corresponding mean altitude resolution of the CO profiles, derived from the FWHM of the averaging kernels.

[Figure]

**Figure 2: Estimated profiles of error for the KIMRA CO profiles. Section 2.4 describes the sources of uncertainties used to estimate the errors. The error due to spectrum noise is an average value over all measurements, and the other errors are calculated about the KIMRA CO a priori profile.**

[Figure]

[Figure]

**Figure 3: The distributions of KIMRA CO partial (40-60 km and 60-80 km) column concentrations divided into groups defined by their position relative to the polar vortex edge using sPV (see Section 3.1). The relative positions are calculated using sPV values at 40, 50, and 60 km, as indicted on respective plots. Using sPV at 50 km gives the three most distinct distributions, as calculated with an unpaired two-sample t-test (See Section 3.1). The \* symbols indicate plots with three distinct distributions.**

[Figure]

**Figure 4: The mean of the coincident MLS CO profiles and the a priori CO profile used for the KIMRA inversion (Section 2.2). MLS orig. shows the mean of the supplied MLS profiles, which use a constant CO concentration above 0.001 hPa (~ 98 km). MLS extrap. shows the mean of the profiles that have been linearly extrapolated in pressure space above 0.001 hPa to provide more physical CO concentrations at these altitudes.**

[Figure]

**Figure 5: (a) The mean of coincident KIMRA and MLS CO profiles from 2011 to 2014, including the mean of the unsmoothed MLS profiles and the a priori profile used for the KIMRA inversion. n is the number of pairs of collocated profiles. (b) The mean of the difference between the KIMRA and smoothed MLS profiles with the standard deviation of the differences as the whiskers on the line. (c) The same as (b) but in relative units, as percent of the mean of KIMRA and MLS CO profiles. (d) The correlation coefficients of KIMRA and smoothed (solid) and unsmoothed (dashed) MLS data. (e) The slope and standard error of a line of best fit to KIMRA vs. smoothed MLS, calculated at each level using given MLS error estimates and two estimations of KIMRA error: the measurement error (black) and double the measurement error (grey) (see Section 3.2). The slope values at 82 and 84 km are unreliable as MLS precision is often quoted as negative at these altitudes. (f) The location of the MLS measurements (magenta) with respect to Kiruna (blue), and a histogram of the times between coincident measurements. The temperature input for the KIMRA inversions shown here includes SSMIS data (see Section 2.2).**

[Figure]

**Figure 6: (a) The absolute mean of the difference in the temperature profiles (ECMWF/MSIS minus NCEP/SSMIS/MSIS) used in the two inversion setups for KIMRA. The altitude ranges of the temperature information used in the profiles (see Section 2.2) are shown here. (b) The mean of the percentage difference in the profiles (difference divided by the average of the two profiles). (c) The correlation between the two datasets. (d) The slope of a line of best fit to the datasets at each level, with ECMWF/MSIS as the dependant (Y) variable. The slope value at the highest altitude shown has a relatively large standard error because of the lower number of points at this altitude after conversion from a pressure to an altitude grid.**

[Figure]

**Figure 7: (a-d) The same calculations as shown in Figure 6 but for the CO volume mixing ratios retrieved using the respective temperature datasets. The slopes and their standard errors (d) are calculated with the same two KIMRA error estimates as in Figure 4 (see Section 3.2), with the larger error bars corresponding to the larger error estimate. Note the different altitude range compared to Figure 6.**

[Figure]

**Figure 8: Daily averaged CO volume mixing ratios (in ppmv) above Kiruna from December 2008 through May 2015. Blank areas within this time are gaps in the data record. Data is plotted using the Isoluminant colormap from Kindlmann et al. (2002), and non-uniformly spaced contours (black lines) between 0.4 and 28 ppmv are added to guide the eye. The temperature input for the KIMRA inversions shown here includes ECMWF analysis (see Section 2.2).**

---

## Author Comment (AC2) · 24 Dec 2016

**Response to comments from Reviewer #2**

This manuscript presents a mesospheric CO dataset covering the period 2008-2015 obtained by means of a ground-based radiometer, KIMRA, installed at Kiruna. The dataset is very valuable and I recommend its publication on ESSD. There are, however, several points I suggest the authors should address before the manuscript is published.

**General comments:**

5

- 1) This manuscript relies on a previous publication (Hoffmann et al., 2011) for what concerns the error analysis description, but the two datasets (the one published in 2011 and the one described here) have important differences (versions of ARTS and QPACK, temperature and pressure profiles, time interval of data used in the comparison, resulting vertical profiles, and more). A description of the uncertainties characterizing the presented dataset should therefore
- 15 be added in this manuscript, also to clarify more precisely which uncertainties and general data characteristics described in Hoffmann et al. apply here as well. This is not clear at the moment. The reference to the discussion of the averaging kernels given in Hoffman et al. (2011) has been expanded to include the main points made in that work that also apply here (Section 2.3): "The centres of the averaging kernels, when represented in VMR, are shifted down in altitude
- 20 compared to a representation using relative concentrations. Hoffmann et al. (2011) provides a detailed discussion of the representation of averaging kernels for ground-based CO measurements, and the major points are worth repeating here. While here the lower limits of the retrievable altitude ranges are set by the SNR of the measurement, the upper limit of the measurements is set by the transition from a pressure broadening regime to a Doppler
- 25 broadening regime (not considering spectrometer channel resolution). The result of this is that, above approximately 70 km in the VMR representation, the altitude locations of the centres of the averaging kernels do not increase anymore with the increase in respective retrieval altitude. And while the retrieved CO values above 70 km do contain information from the atmosphere corresponding to the retrieval altitude, the VMR representation of the data above this altitude
- 30 should be considered with care."

Two new sections have been added to address the errors in the profile and a Figure has been added to plot the respective error contributions to the CO profiles.

**"2.4 Error estimates for the retrieved dataset**

- 35 Errors in the retrieved CO profiles arise from uncertainties in instrumental parameters and in parameters that are used as input to the forward model. The relative contributions from these uncertainties are calculated here using the OEM error definitions and by introducing perturbations to the inputs. OEM error definitions are described in detail in Rodgers (2000), among others, and are not repeated here. Figure 2 shows the estimated contributions to the CO
- 40 profile error budget from the following uncertainties. There are, of course, possible sources of error in any stage of the instrumentation, and the parameters discussed here are considered to represent the dominant uncertainties.

The statistical noise,  $\Delta T$ , on a spectrum is governed by the so-called radiometer equation  $\Delta T = T_{sys}/\sqrt{\nu\tau}$ , with  $T_{sys}$  as the system noise temperature of the instrument,  $\Delta v$  as the channel bandwidth (Table 1), and  $\tau$  as the integration time for a measurement of the atmospheric signal. The uncertainty for the temperature profile used in an inversion is the same uncertainty used in

- 5 Hoffman et al. (2011): 5% below 80 km, 10% above 100 km, and linearly interpolated in between. Uncertainties in the spectroscopic parameters of the CO line are from the HITRAN 2008 catalogue and are: 1% for the line intensity, 2% for the air broadening parameter, and 5% for the temperature dependence of air broadening. Uncertainty in the line position is ignored as the frequency grid can be adjusted to match the line centre in a measurement, and the parameters
- 10 associated with self-broadening are considered as having negligible impact due to the relatively low concentration of the observed gas (Ryan and Walker, 2015). For the blackbody targets used in calibration of the atmospheric measurement, an uncertainty of 2 K is assumed in the temperature of the hot and cold target: a conservative estimate that accounts for fluctuations and drifts in temperature. The uncertainty in the pointing of the instrument to the sky is estimated
- as 1°. This is an order of magnitude higher than the precision of the motor that controls pointing, to allow for a possible offset in the orientation of KIMRA.
   The resulting error estimates as well as a sum, in quadrature, of the errors are plotted in Figure 2, and show that the predominant error in the CO profiles comes from the statistical measurement noise on the spectrum. This peaks at 30% of the mean KIMRA CO profile, at 53 km altitude. The
- 20 error from uncertainties in the temperature profile also shows a significant contribution to the total error in the retrievable altitude range above approximately 60 km. The error due to noise on the spectrum is calculated during each inversion and is provided in the supplemental data with the corresponding CO profiles. The error profile in Figure 2 is an average over all measurements. The other plotted errors are calculated about the a priori CO profile and serve as an estimate of
- 25 the respective error contributions to each measurement. To calculate these errors for individual measurements is computationally expensive and so the values plotted in Figure 2 are provided in the supplemental data.

**2.5 Smoothing error and interpretation of the KIMRA profiles**

Smoothing error in the profiles arises from the limited vertical resolution of the retrieved profile and can be calculated with the OEM using the averaging kernels and a priori information for a profile. Smoothing error can be large for ground-based profiling instruments due the small altitude spacing between retrieval grid points, chosen here for numerical stability in the inversion (Eriksson, 1999), compared to the actual vertical resolution of the retrieved profiles (see Figure 1 (c) and (d)). Smoothing error should be assessed when one wishes to use or interpret the CO

- 35 profiles without consideration of the accompanying averaging kernels. As this is not a recommended use of the data, the smoothing error is not assessed here. If using KIMRA profiles to say something of the absolute value of CO at a given retrieval altitude, one must be aware that a CO value at a retrieval grid point contains information from a range of altitudes with a sensitivity governed by the shape of the corresponding averaging kernel.
- 40 Smoothing error can be accounted for when comparing KIMRA CO profiles to those of an instrument/model that has a significantly different vertical resolution by using the KIMRA averaging kernel matrices and a priori to smooth (Rodgers and Connor, 2013) the other profiles

so that they have a similar vertical resolution. KIMRA profiles can be used to observe changes in CO concentrations over time, provided there is not a significant difference in the averaging kernels (and thus measurement response) of the profiles over that time. In particular, care should be taken with data near the edges of the retrievable altitude range (where the measurement response is decreasing towards 0.8) as the measurement response in this region can change

- 5 response is decreasing towards 0.8), as the measurement response in this region can change quickly when there are sharp changes in atmospheric CO."
- 2) In order to consider this manuscript a validation of this updated and extended dataset, the authors should be more quantitative on the biases with respect to Aura/MLS and show in more details the differences between KIMRA and MLS. It would be important to know and state what the relative difference between the two datasets is, at all (most) altitudes. In the abstract, authors state that the two sets of data have a "high level of agreement", but a possible bias of
- 15 10-15% over parts of the vertical coverage does not qualify as "high level of agreement". Figure 4 (b) also shows that in the upper mesosphere (65-85 km) there's a large std dev of the difference (which is not sufficiently stressed in the manuscript) and this weakens the stated agreement between the two datasets.

The relative mean and standard deviation of the differences has been added to the Figure comparing KIMRA and MLS. The text in Section 3.2 has been modified to offer some quantitative values of the comparison.

"The mean profiles show a maximum absolute bias of ~0.65 ppmv at 68 km, and a maximum relative bias of 22% (0.44 ppmv) at 60 km, with bias being defined as KIMRA minus MLS. The standard deviation of the differences in the profiles peaks at 21% at 60 km. These standard deviation values are similar in magnitude to the estimated uncertainties in the KIMRA CO profiles.

25 deviation values are similar in magnitude to the estimated uncertainties in the KIMRA CO profiles (Figure 2)."

The abstract has been edited and now reads, of the MLS and KIMRA comparison:

- "This KIMRA CO dataset is compared with CO data from Microwave Limb Sounder aboard the
  Aura satellite: There is a maximum bias for KIMRA of ~ 0.65 ppmv at 68 km (corresponding to
  14.7% of the mean CO value at 68 km), and a maximum relative bias of 22% (0.44 ppmv) at 60 km. Standard deviations of the differences between profiles are similar in magnitude to the estimated uncertainties in the profiles. Correlations between the instruments are within 0.87 and
  0.94. These numbers indicate agreement between the instruments."
- 35

**The end of Section 3.2 has been edited to:**

"Despite the > 1.0 slope values, KIMRA and MLS are considered to show agreement, according to the level of difference and correlation between profiles."

40

3) The uncertainties and potential biases of the dataset uploaded in the database are not clear. I agree that it is useful to underline how important it is the use of different T and P vertical profiles in the analysis (and hence the presented study on the two different KIMRA set of retrievals), but eventually potential users/readers are interested in knowing how to characterize/treat/use the data made available, e.g., the retrievals associated to the ECMWF reanalysis. As a start, in the various comparisons discussed in the manuscript the authors should always state if the bias is positive or negative, i.e., which dataset is high or low.

5

**Section 1 has been modified to clarify bias:**

"The comparisons showed good agreement below 60 km but at higher altitudes the profiles significantly diverged leading to a positive bias in KIMRA CO of > 5 ppmv at 80 km. The shape of the bias in the profile was consistent between satellite datasets and its origin is unclear."

10

Section 3.2 has been modified to clarify that the bias refers to KIMRA minus MLS. "The mean profiles show a maximum absolute bias of ~0.65 ppmv at 68 km, and a maximum relative bias of 22% (0.44 ppmv) at 60 km, with bias being defined as KIMRA minus MLS."

15 Section 4.1 defines the comparison as: "(ECMWF/MSIS minus/vs. NCEP/SSMIS/MSIS)".

For the error estimates in the profile and use of the data please refer to the answer to General Comment 1.

20

4) The altitude range of the dataset that is recommended for scientific use is not clear, as numbers change in the manuscript and differ from the 40-80 km range (with the interval 70-80 km to be used with care) recommended in Hoffmann et al.

Hoffman et al. (2011) states that the retrieval

25 "works generally reasonable between 40 and 80 km with a vertical resolution of 16 to 22 km. However, the region between 70 and 80 km has to be treated with care depending on the particular application.".

And also states

- "Assuming that an AVK area greater than 0.8 contains enough information from the measurement, we find general sensitivity (Fig. 4a) in a range of 34 to 87 km for the retrieval converted to vmr and a range of approx. 27 to 83 km for the fractional retrieval. This range matches the maximum expectations, but is narrowed by considering further criteria in the following.".
- 35 Section 2.3 in the current manuscript states that the average retrievable altitude range is 48 84 km and has been edited to include the main points in the discussion of the averaging kernels in Hoffman et al. (2011), namely that the data above 70 km should be treated carefully (see the response to General Comment 1). Specifically, the part of Section 2.3 now reads:
- 40 *"The CO profiles here have an average retrievable altitude range of 48–84 km and an average vertical resolution of between 15 and 18.5 km depending on the altitude. These compare to an average altitude range of 40-80 km and average resolution of 16-22 km for the CO profiles shown in Hoffmann et al. (2011) for the winters of 08/09 and 09/10. DOFS for the current dataset have a mean of 2.0 and a standard deviation of 0.6. The minimum and maximum of the lower and*

upper retrieval limit for the current dataset is 35 km and 99 km, respectively, but the upper limit of any profile (defined using measurement response) must be considered with the following caveat.

- The centres of the averaging kernels, when represented in VMR, are shifted down in altitude compared to a representation using relative concentrations. Hoffmann et al. (2011) provides a detailed discussion of the representation of averaging kernels for ground-based CO measurements, and the major points are worth repeating here. While here the lower limits of the retrievable altitude ranges are set by the SNR of the measurement, the upper limit of the measurements is set by the transition from a pressure broadening regime to a Doppler
- 10 broadening regime (not considering spectrometer channel resolution). The result of this is that, above approximately 70 km in the VMR representation, the altitude locations of the centres of the averaging kernels do not increase anymore with the increase in respective retrieval altitude. And while the retrieved CO values above 70 km do contain information from the atmosphere corresponding to the retrieval altitude, the VMR representation of the data above this altitude.
- 15 should be considered with care."

5) Hoffmann et al. (2011) explicilly stated that the use for that dataset was only for comparison and validation with higher vertical resolution datasets, and therefore the error associated with

20 those profiles did not include, for example, the smoothing error. I personally find this approach correct. Here, however, it is a different matter. Authors should advice potential users that intend to employ the updated and extended KIMRA dataset as a stand alone product that the uncertainties in the mixing ratio profiles can be much larger due to, for example, the smoothing error. More importantly, I think that in the dataset there should be the information on the

**25 estimated total uncertainty, as it is described for example by Hoffmann et al. Hoffman et al. (2011) states: "The main application of the presented retrieved dataset is the comparison to modeled data or satellite observations with a focus on the temporal CO variability."**

And also (in the same section):

- 30 "The characteristics for this vmr representation are therefore not ideal. This is, however, only relevant if the vmr itself is of particular interest (in contrast to e.g. its variation in time) and individual KIMRA vmr profiles are regarded as stand-alone and not relative to each other or to independent data. Since such a stand-alone use is not a major application of the presented dataset, the presented KIMRA retrieval is not optimized for this case."
- 35 A specific point is made that *"stand alone"* refers to *"the vmr itself"* and not to *"its variation time", for example.* Hoffmann et al. (2011) also go on to use the CO dataset to estimate the average descent velocity above Kiruna.

Regarding error estimation, this is now included as Section 2.4 "Error estimates for the retrieved
40 dataset". Please see the response to General Comment 1.

Smoothing error is not included in the error estimates for the profile for reasons similar to those given by Hoffman et al. (2011), among others. This information and some more elaboration is

now included as Section 2.5 "Smoothing error and interpretation of the KIMRA profiles". Please see the response to General Comment 1.

5

**Specific comments:**

Page 4, Lines 1-2: Is the sinewave removal accounted for in the error estimate and sensitivity of the retrieval? Sinewaves should also be shown in Figure 1 to give the reader an idea of what their amplitude might be compared to the errors listed and the spectral line intensity.

- their amplitude might be compared to the errors listed and the spectral line intensity.
   Page 4, Lines 1-2: Same as above.
   Section 2.2 has been edited to include more information on the fit to the baseline, which is included in the inversion (i.e., the optimal estimation).
   *"A fitting of functions to the baseline of the measurement (baseline fit) can be included in the*
- 15 optimal estimation performed by Qpack 2 and forms part of the general fit to the measurement (inversion fit). For sine waves, the period and estimated amplitude uncertainty are provided as input, and the amplitude and phase of the waves are retrieved. The periods of the sine waves in the KIMRA spectra were found by first inverting the all of the measurements without a fit to the baseline and then evaluating a periodogram of the residuals. This procedure was also applied to
- 20 subsets of the data in case some changes in the estimated periods with time became evident. Sine waves in the baseline are not generally visible by eye on the CO spectra but there were no changes over time found in the determined wave periods. The periods of the fitted waves are 27.5 MHz, 55 MHz, and 36.3 MHz, with an estimated uncertainty in the amplitudes of 0.5 K, 0.3 K, and 0.5 K, respectively. These calculated sinewave periods are almost identical those found by Hoffman
- 25 et al. (2011) for the 08/09 and 09/10 winters (27.5 MHz, 55 MHz, and 36.6 MHz), and are similarly large in comparison to the width of the CO spectral line, so that they are uniquely distinguishable from it. A second order polynomial is also fitted to the baseline during the optimal estimation to account for any offsets or long-period sine wave signatures; The zeroth, first, and second order coefficients have estimated uncertainties of 1 K, 0.5 K, and 0.5 K, respectively."
- 30

The fitted baseline is now included alongside the residual in Figure 1. Some extra information is provided on the standard deviation of the retrieved baseline in comparison to the standard deviation of the statistical noise on the spectrum.

*"Figure 1 (b) shows an example spectrum of a CO measurement with KIMRA and the corresponding inversion fit. The inversion fit includes the baseline fit described in Section 2.2 and the example baseline fit is plotted alongside the residual in Figure 1 for comparison. Considering the entire dataset, the standard deviation (averaged across all spectrometer channels) of the amplitude of the fitted baseline is 0.21 K, and the average standard deviation of the residual is 0.34 K. In other words, changes in the retrieved amplitude of the baseline are, on average, lower*

40 than the statistical measurement noise on the spectrum."

**Page 4, Lines 15: Co-authors should not be cited in personal communication, I think.**

This is the most accurate description and editing it would give a distorted view of events.

**Page 4, Lines 27: See general comment #1.**

This line has been removed. Please see the response to General Comment 1.

5 Page 5, Line 10: Looking at the dataset, there are profiles which have a measurement response larger than 0.8 at 93 km altitude. This upper limit for the retrieval is, however, unrealistic. Authors should state explicitly in the manuscript and in the uploaded dataset which is the altitude range where KIMRA profiles are reliable.

Please see the response to General Comment 4.

10

**Page 5, Line 17: See general comment #1.**

Noted. Please see the response to General Comment 1.

15

Page 5, Line 20: This concept is clear. However, you should also advice potential users that intend to employ your dataset as a stand alone product.

Information on this is now included in Section 2.5 *"Smoothing error and interpretation of the KIMRA profiles"*. Please also see the response to General Comment 5.

20

25

**Page 5, Line 31: figure 4e seems to suggest a larger coincidence criteria.**

Section 3.1 now includes "The location of a measurement, its position relative to the vortex, and the distance between measurements was calculated at 50 km altitude." to clarify that the positions are calculated at 50 km altitude. With KIMRA observing at elevation angles less than 90 degrees, the point of colocation may be farther than 2 degrees latitude from Kiruna.

**Page 6, Line 15: In the X-title of the plots of figure 2 it says 46-66 km, instead of 40-60. Which is it?**

30 The error was in the text. This has been fixed to 46-66 km and 66-86 km.

**Page 7, Line 4: I am not sure this figure is necessary.**

A significant change is made to the provided MLS CO profile and the realization of the change should be made clear to readers. From this point, the figure should be retained in the manuscript.

35

**Page 7, Line 10: Please, explain in more details what is this correlation. If you refer exactly to what has been done by Hoffmann et al., please say so.**

This is clarified as being the sample Pearson correlation coefficient.

"The mean of KIMRA and MLS CO profiles are shown (a) as well as the mean of the differences
(bias) in the coincident profiles in ppmv (b) and relative to the mean of KIMRA and MLS profiles
(c), the sample Pearson correlation coefficient for the profiles at each altitude (d), and the slope of a line of best fit to KIMRA vs. MLS at each altitude layer (e), explained below."

Page 7, Line 14: In figure 4, the difference between MLS and MLS smooth is not convincing. Kimra averaging kernels are apparently pushing MLS towards larger CO vmr values with respect to its original values. A good set of AKs should just smooth the original dataset, degrade it, not add positive or negative biases. Furthermore, this plot doesn't illustrate what the actual

**5 differences between KIMRA and MLS is. X-scale is maybe too large, and it would be useful also to see the relative difference in percentage.**

The x-scale has been shortened and the relative difference is now included in the figure.

In Figure 4 (a): Below 60km, the smoothed profile is larger than the original. Between 60 km and

- 10 65 km, the smoothed profile is smaller than the original profile. Between 65 km and 83 km, the smoothed profile is larger than the original. Above 80 km, the smoothed profile is less than the original profile. It is understandably hard to see this in the figure. The line types have been changed to make this clearer in the figure and the relative difference is also now shown.
- Such variations in a smoothed and original profile are common. One example can be seen for an
   ozone comparison with a ground-based MWR satellite-borne SBUV/2 in Ohyama et al. (2016),
   Figure 8.

Figure 4 (a) here also shows average profiles, in which short scale variations are not usually visible.

The smoothing of a profile degrades the vertical resolution of the satellite profile, but is more complex because it also adds information from the ground-based a priori profile and information

20 complex because it also adds information from the ground-based a priori profile and information about the shape and altitude sensitivity of the averaging kernels.

Page 7, Line 20: I understand that this analysis replaces the older one, but maybe a tentative guess as to what is working now that wasn't working before should be attempted. Also, if the analyses are so different, more needs to be said concerning the error analysis of this latest one (see general comment #1).

Regarding the error analysis, please see the response to General Comment 1.

Regarding a tentative guess, the authors would prefer not to guess or speculate on the cause of
the difference. An in-depth comparison is not the aim of the work and falls outside of the scope of this contribution.

Page 7, Line 28: This is true only up to 65 km, above that (a large fraction of the profile) Hoffmann et al. show that the predominant error is due to the doppler broedening, e.g.,

35 temperature, and in this case doubling the measurement error is definitely not an overestimation, but again an underestimation.

Information is now added in the error analysis section (Section 2.4) and shows that the predominant error in the dataset is due to measurement noise in the spectrum. Please see the response to Section 1 and Figure 2 of the edited manuscript.

40

**Page 7, Line 29: It's not clear to me why a slope greater than one means greater variation in KIMRA CO concentrations with respect to MLS. Please, explain.**

This has been changed to say "...KIMRA shows a greater range of CO concentrations...".

A greater range in the Y-axis compared to the X-axis provides a slope greater than 1. Similar conclusions are drawn in Nedoluha JGR VOL. 102, NO. D14, PAGES 16,647-16,661, JULY 27, 1997, for water vapour comparisons with WVMS and HALOE.

**5**

Page 8, Line 2: In order to consider this manuscript a validation of this dataset, the authors should be more quantitative and show in more details the differences between KIMRA and MLS (see general comment #2).

This sentence now reads "Despite the > 1.0 slope values, KIMRA and MLS are considered to show agreement, according to the level of difference and correlation between profiles."

Please also see the response to General Comment 2.

**Page 8, Line 11: This sentence is unclear.**

15 This sentence has been edited to:

"A comparison of the two sets of temperature profiles (ECMWF/MSIS minus/vs. NCEP/SSMIS/MSIS) is shown in Figure 6. (a) and (b) show the mean and standard deviation of the differences between temperature profiles in absolute and relative units, (c) shows the correlation at each altitude, and (d) shows the slope of the lines of best fit at each altitude."

20

Page 8, Line 21: I understand that MSIS profiles from a different time of the day can be different from one another. Yet, looking at figure 5, their difference at 115 km and their low correlation at 100 km are surprising.

- 25 The MSIS data was checked and a change of 4% (about 10 K) at 115 km appears to be common. There is often a change in slope of the temperature profile that occurs around 100 km in the MSIS profiles, which may be causing the drop in correlation between nearby (in time) profiles. As the main aim here is to compare the different temperature datasets and their effect on the retrieved KIMRA CO profiles, it is outside the scope of the work to perform a detailed study of
- 30 the MSIS temperature profiles.

**Page 9, Line 10: By using your dataset only, how can you distinguish a SSW from a simple shift of the vortex further away from kiruna?**

- It is known through the referenced material that SSWs occurred at these dates and the effects on CO concentrations are known, also from the referenced material. This can be seen in the KIMRA data and is commented on here. It is made clear at the beginning of the paragraph that one cannot use the data alone to make such a characterization: *"While it is impossible to fully characterise the concentrations shown without inclusion of other instrument data and/or model output, some observations are made here."*
- 40

**Page 9, line 13: Remove point after "2015)".**

Thank you, this has been fixed.

**Page 9, Line 19: As commented earlier, what about users of this dataset as a stand alone? Should you add to the dataset also a column with the total uncertainty?**

Please see the response to General Comment 5, regarding use of the dataset as stand-alone. The individual errors plotted in Figure 2 will now be added to the data files and resubmitted to the PANGAEA database

5 the PANGAEA database

Page 9, Line 23: Is this the official altitude range? What about the considerations in Hoffman et al. that limit the upper altitude to 80 km? Also, in figure 4 (b), are the ï A DVMR below 50 km particularly small because the inversions don't have enough sensitivity below 50 km?

- 10 This information has been edited to say: *"The resulting profiles cover an average altitude range of 48-84 km. The retrievable altitude limits vary with the SNR of each measurement, and CO VMRs above 70 km should be treated with care due an offset in centres of the corresponding averaging kernels above this altitude."*
- 15 The main reason for a low absolute difference below 50 km is that the CO values at these altitudes are generally low. Figure 4 (5 in the edited manuscript) now includes the differences in relative units. The part of the conclusion referring to this now reads: *"The instruments show agreement, with KIMRA showing a maximum bias of ~ 0.65 ppmv at 68 km (corresponding to 14.7% of the mean CO value at 68 km), and a maximum relative bias of 22%*
- 20 (0.44 ppmv) at 60 km. Correlations with MLS are greater than 0.90 at most altitudes.."

References: Straub et al., 2013, is missing.

This has been added

25

Figure 2: Units of the X-axis are missing.

This has been fixed.

**Strato-mesospheric carbon monoxide profiles above Kiruna, Sweden (67.8° N, 20.4° E), since 2008**

Niall J. Ryan1, Mathias Palm1, Uwe Raffalski2, Richard Larsson3, Gloria Manney4, Luis Millán5, Justus Notholt1

1Institute of Environmental Physics, University of Bremen, Bremen, 28359, Germany
 2Swedish Institute of Space Physics, Box 812, SE-981 28 Kiruna, Sweden
 3National Institute of Information and Communications Technology, Tokyo, Japan
 4NorthWest Research Associates, Socorro, New Mexico, USA and Department of Physics, New Mexico Institute of Mining and Technology, Socorro, New Mexico 87801, USA

5Jet Propulsion Laboratory, M/S 183-701, 4800 Oak Grove Drive, Pasadena, CA 91109, USA

Correspondence to: Niall J. Ryan (ryan.niall@gmail.com)

Abstract. This paper presents the retrieval and validation of a self-consistent timeseries of carbon monoxide (CO) above Kiruna using measurements from the Kiruna Microwave Radiometer (KIMRA). The dataset currently spans
the years 2008 to 2015, and measurements are ongoing at Kiruna. The spectra are inverted using an optimal estimation method (OEM) to retrieve altitude profiles of CO concentrations in the atmosphere within an average altitude range of 48 - 84 km. Atmospheric temperature data from the Special Sensor Microwave Imager/Sounder aboard the US Air Force meteorological satellite, DMSP-F18, are used in the inversion of KIMRA spectra between January 2011 and May 2014. This KIMRA CO dataset is compared with CO data from Microwave Limb Sounder aboard the Aura

- 20 satellite: There is a maximum bias for KIMRA of ~ 0.65 ppmv at 68 km (corresponding to 14.7% of the mean CO value at 68 km), and a maximum relative bias of 22% (0.44 ppmv) at 60 km. Standard deviations of the differences between profiles are similar in magnitude to the estimated uncertainties in the profiles. Correlations between the instruments are within 0.87 and 0.94. These numbers indicate agreement between the instruments. To expand the CO dataset outside of the lifetime of DMSP-F18, another inversion setup was used that incorporates modelled
- 25 temperatures from the European Centre for Medium-Range Weather Forecasts. The effect on the retrieved CO profiles when using a different temperature dataset in the inversion was assessed. A comparison of the two overlapping KIMRA CO datasets shows a positive bias of < 5% in the extended dataset and a correlation > 0.98 between the lower retrievable altitude limit and 82.5 km. The extended dataset shows a larger range (≤ 6%) of CO concentrations that is not explained by random error estimates. Measurements are continuing and the extended KIMRA CO timeseries
- 30 currently spans 2008 to 2015, with gaps corresponding to non-operation and summer periods when CO concentrations below ~ 90 km drop to very low values.

The data can be accessed at: https://doi.pangaea.de/10.1594/PANGAEA.861730.

**1** Introduction**

35

The principle source of carbon monoxide (CO) in the middle atmosphere is the photolysis of carbon dioxide (CO2) in the thermosphere and its subsequent vertical transport, and its only sink is through reaction with the hydroxyl radical ('OH) (Solomon et al., 1985). The loss rates in the thermosphere are low and this leads to a strong vertical gradient in CO concentrations (volume mixing ratio (VMR) is the form of gas concentration used throughout this work, and the two terms are considered to be synonymous here). As the production and loss mechanisms for atmospheric CO require the presence of sunlight, the lifetime of CO during polar winter is on the order of months (Solomon et al., 1985, Allen et al., 1999), making it an excellent tracer for atmospheric dynamics. In spring the lifetime in the upper stratosphere can be 15 - 20 days poleward of  $60^{\circ}$  latitude (Minschwaner et al., 2010). Due to the longer CO lifetimes within the

- 5 polar vortex during winter, there exists also a strong horizontal concentration gradient across the vortex boundary. While satellite measurements of CO profiles have been used regularly to study atmospheric transport processes, particularly during northern winters (e.g. Damiani et al., 2014, Lee et al., 2011, Manney et al., 2009, McLandress at al., 2013), ground-based CO profile datasets for the poles are few and far between. The Ground-Based Millimeter-Wave Spectrometer (GBMS) installed in Thule Air Base, Greenland (76.5°N, 68.7°W), has been used to study the
- 10 composition of Arctic winter atmosphere in 2001/2002 (Muscari et al., 2007) and the sudden stratospheric warming (SSW) of 2009 (Biagio., et al., 2010). The Onsala Space Observatory instrument (57°N, 12°E) has measured CO in 2002-2008 (Forkman et al., 2012), and from 2014. The British Antarctic Survey (BAS) Radiometer dataset for Troll Station (72°S, 2.5°E) covers February 2008 to January 2010 (Straub et al., 2013). Each of these three ground-based instruments are microwave radiometers that measure emissions from molecules undergoing rotational transitions in
- 15 the atmosphere, offering the advantage of providing measurements during polar night, compared to instruments that rely on the sun.

This paper presents a CO profile dataset from 2008 to 2015 from measurements made by the Kiruna Microwave Radiometer (KIMRA) at the Swedish Institute for space Physics, Kiruna (67.84°N, 20.41°E). KIMRA measurements during the winters 2008/2009 and 2009/2010 have previously been used (Hoffmann et al., 2011) to retrieve CO profiles

20 that have been compared to satellite data from: the Microwave Limb Sounder (MLS) aboard Aura, the Atmospheric Chemistry Experiment - Fourier Transform Spectrometer (ACE-FTS) aboard SCISAT-1, and the Michelson Interferometer for Passive Atmospheric Sounding (MIPAS) aboard ENVISAT. The comparisons showed good agreement below 60 km but at higher altitudes the profiles significantly diverged leading to a positive bias in KIMRA CO of > 5 ppmv at 80 km. The shape of the bias in the profile was consistent between satellite datasets and its origin

is unclear.

This paper presents a CO dataset from KIMRA, using an extended set of measurements, beginning in 2008, and retrieved using a new inversion setup. The layout of the paper is as follows: Section 2 provides details on the KIMRA observation system: the measurements and the inversion technique. Section 2 also offers an estimation of the errors in CO profiles, and a word on interpretation of the data. Section 3, to establish validity of the observation system, presents

- 30 a comparison of the KIMRA data with MLS data, using temperature information from the Special Sensor Microwave Imager/Sounder (SSMIS) (Kunkee et al.,2008; Swadely et al., 2008) aboard the US Air Force's Defense Meteorological Satellite Program F-18 satellite as input for the KIMRA CO inversion. Section 4 investigates changes in the retrieved KIMRA CO data when using temperature information from the European Centre for Medium-Range Weather Forecasting (ECMWF), and presents this extended dataset, currently spanning 2008-2015. Section 5 concerns
- 35 the availability and use of the data, and Section 6 offers some concluding remarks. Satellite data is exchanged for ECMWF analysis in the KIMRA inversion in order to have a consistent temperature input for the entire KIMRA dataset; The timespan of continuing KIMRA measurements will generally surpass the

lifetime of any satellite instrument. Many satellite instruments have well exceeded their mission lifetime, the three instruments mentioned in this section being good examples: MLS (2004 - present), ACE-FTS (2003 - present), and MIPAS (2002 - 2012). Ground-based instruments, however, have the potential to produce much longer datasets, albeit at one location, due to the much smaller cost of the ground-based projects and the ability to maintain the instruments.

- 5 The ground-based radiometer OZORAM has been measuring ozone in the Arctic (79.9° N, 11.9° E) since 1994 (Palm et al., 2010), and Nedoluha et al. (2016) recently showed twenty years of chlorine monoxide measurements in the Antarctic (77.85° S, 166.77° E) with the ground-based radiometer ChlOE1. The Fourier transform infrared spectrometer at Kitt Peak (31.9° N, 111.6° W) (Brault et al., 1978) produces datasets beginning in the 1970's. These are just two examples of a large number of ground-based instruments. Long-term (decades) ground-based datasets are
- 10 important as they reveal changes in the atmosphere on the time scale of changes in Earth's climate, and the data can be used to validate satellite instruments, fill gaps in time between satellite missions, and to help combine satellite datasets that do not overlap in time.

**2 Instrument and dataset**

**2.1 KIMRA**

- 15 KIMRA is housed at the Swedish Institute for Space Physics (IRF), Kiruna, and was partly designed by the Institute for Meteorology and Climate Research (IMK) at the Karlsruhe Institute of Technology (KIT) (Raffalski et al., 2002). KIMRA utilises the frequency range of 195 233 GHz and has been measuring, among others, atmospheric spectra that correspond to the J = 2 → 1 rotational transition (230.54 GHz) of CO. KIMRA has operated in Kiruna since 2002 and has been making measurements of CO emissions since 2008. The more general aspects of the instrument are given
- 20 in Table 1 and the more specific details can be found in Raffalski et al. (2002) and Hoffmann et al. (2011). The spectrometer used for CO measurements is a Fast Fourier Transform Spectrometer (FFTS) made by Omnisys Instruments, with 1024 channels and used with a bandwidth of 110 MHz to give a resolution of ~107 kHz per channel. For a given measurement cycle, which includes multiple hot/cold calibrations and produces one time-averaged atmospheric spectrum, KIMRA points to the atmosphere at an elevation angle, between 5° and 90°, that is chosen to
- 25 provide the best signal-to-noise ratio (SNR) at that time. This angle is governed by the atmospheric conditions and so can change from one measurement cycle to another. Because they are produced using different elevation angles, the individual spectra are not averaged to reduce the SNR. Spectral averaging has been used for similar measurements from other instruments that vary the elevation angle, e.g., over timescales of a week in Nedoluha et al. (2013) and an hour in Ohyama et al. (2016). Rather, each KIMRA spectrum is used to retrieve a CO profile, which may then be used
- 30 in an average. The azimuth angle of a KIMRA measurement is either 0° (directly north) or 180° (directly south) and can change from one measurement cycle to another. The azimuth and elevation angle are kept constant during each measurement cycle. Figure 1 (a) shows the distribution of the duration times of each measurement.

**2.2 Inversion setup**

CO profiles are retrieved from the spectra using an optimal estimation method (OEM) inversion (Rodgers, 2000). This is a Bayesian statistical approach that constrains the retrieved CO profile according to some a priori atmospheric information. The inversion was carried out using the Qpack 2 (Eriksson et al., 2005) package, which employs the

- 5 Atmospheric Radiative Transfer Simulator: ARTS 2 (Eriksson et al., 2011) to model radiative transfer through the atmosphere, i.e., the forward model. All of the following information that is input into the inversion is done so using Qpack 2. The a priori CO information used here is the average of one winter period (September through April) of output from the Whole Atmosphere Community Climate Model, version 4 (WACCM4) (Garcia et al., 2007) provided by Douglas Kinnison at the National Center for Atmospheric Research (NCAR), with a standard deviation of 100%
- 10 at all altitudes. This combination was found to give enough freedom to the inversion to fit expected changes in CO above Kiruna throughout a given winter period (here September through May) while providing enough regularisation of the retrieved solution so that no oscillations are readily observed in the CO profiles. The WACCM data is on a 132-layer grid between approximately ground and 130 km. Ozone (O3) is also retrieved simultaneously with CO, as there is an O3 spectral line located at 231.28 GHz, and attenuation of the CO spectral line due to water vapour is accounted
- 15 for by including the water vapour continuum described by Rosenkranz (1998) in the forward model and inversion. The spectroscopic information used is from the HITRAN 2008 catalogue (Rothmann et al., 2009). Continua of molecular oxygen (O2) and nitrogen (N2) (Rosenkranz, 1993), and nitric acid (HNO3) lines are also included in the inversion but are not retrieved and are considered model parameters. A priori profiles of O3, water vapour, and O2 are from the same WACCM run as the CO a priori, and N2 and HNO3 a priori profiles are from the FASCOD subarctic

winter scenario (Anderson et al., 1986). A priori profiles of CO, O3, water vapour, O2, and N2 are unchanged from those described in Hoffman et al. (2011 and 2012).
 Measurement noise (statistical noise on the spectrum) was estimated by fitting a second order polynomial to a wing of the spectrum and calculating the standard deviation of the fit. As there is no windowing applied in the operation of

the FFTS, the spectrometer channels are specified in the inversion as having a sinc-squared response function (Harris,

- 25 1978). Three sine wave functions are fitted to the baseline of each spectrum during an inversion to account for errors in the baseline, which are most often produced by standing waves in the instrument. A fitting of functions to the baseline of the measurement (baseline fit) can be included in the optimal estimation performed by Qpack 2 and forms part of the general fit to the measurement (inversion fit). For sine waves, the period and estimated amplitude uncertainty are provided as input, and the amplitude and phase of the waves are retrieved. The periods of the sine
- 30 waves in the KIMRA spectra were found by first inverting all of the measurements without a fit to the baseline and then evaluating a periodogram of the residuals. This procedure was also applied to subsets of the data in case some changes in the estimated periods with time became evident. Sine waves in the baseline are not generally visible by eye on the CO spectra but there were no changes over time found in the determined wave periods. The periods of the fitted waves are 27.5 MHz, 55 MHz, and 36.3 MHz, with an estimated uncertainty in the amplitudes of 0.5 K, 0.3 K,
- 35 and 0.5 K, respectively. These calculated sinewave periods are almost identical to those found by Hoffman et al. (2011) for the 08/09 and 09/10 winters (27.5 MHz, 55 MHz, and 36.6 MHz), and are similarly large in comparison to the width of the CO spectral line, so that they are uniquely distinguishable from it. A second order polynomial is also

[revised manuscript text omitted]

Figure 1 (b) shows an example spectrum of a CO measurement with KIMRA and the corresponding inversion fit. The inversion fit includes the baseline fit described in Section 2.2 and the example baseline fit is plotted alongside the

- 5 residual in Figure 1 for comparison. Considering the entire dataset, the standard deviation (averaged across all spectrometer channels) of the amplitude of the fitted baseline is 0.21 K, and the average standard deviation of the residual is 0.34 K. In other words, changes in the retrieved amplitude of the baseline are, on average, lower than the statistical measurement noise on the spectrum.
- The mean of the averaging kernels for the CO dataset are also shown in Figure 1 (c) along with the measurement response (d) (sum of the rows of the averaging kernel matrix) and the altitude resolution of the profiles (full-width-at-half-maximum of the averaging kernels). Altitudes with a measurement response greater than 0.8 are considered to represent the range of useful profile information: the retrievable altitude range. This choice is somewhat arbitrary, with 0.8 being a regularly used value (e.g., Forkman et al., 2012; Hoffmann et al., 2011; Straub et al., 2013). The CO profiles here have an average retrievable altitude range of 48–84 km and an average vertical resolution of between 15
- 15 and 18.5 km depending on the altitude. These compare to an average altitude range of 40-80 km and average resolution of 16-22 km for the CO profiles shown in Hoffmann et al. (2011) for the winters of 08/09 and 09/10. DOFS for the current dataset have a mean of 2.0 and a standard deviation of 0.6. The minimum and maximum of the lower and upper retrieval limit for the current dataset is 35 km and 99 km, respectively, but the upper limit of any profile (defined using measurement response) must be considered with the following caveat.
- 20 The centres of the averaging kernels, when represented in VMR, are shifted down in altitude compared to a representation using relative concentrations. Hoffmann et al. (2011) provide a detailed discussion of the representation of averaging kernels for ground-based CO measurements, and the major points are worth repeating here. While here the lower limits of the retrievable altitude ranges are set by the SNR of the measurement, the upper limit of the measurements is set by the transition from a pressure broadening regime to a Doppler broadening regime (not considering spectrometer channel resolution). The result of this is that, above approximately 70 km in the VMR
- representation, the altitude locations of the centres of the averaging kernels do not increase anymore with the increase in respective retrieval altitude. And while the retrieved CO values above 70 km do contain information from the atmosphere corresponding to the retrieval altitude, the VMR representation of the data above this altitude should be considered with care.

**30 2.4 Error estimates for the retrieved dataset**

Errors in the retrieved CO profiles arise from uncertainties in instrumental parameters and in parameters that are used as input to the forward model. The relative contributions from these uncertainties are calculated here using the OEM error definitions and by introducing perturbations to the inputs. OEM error definitions are described in detail in Rodgers (2000), among others, and are not repeated here. Figure 2 shows the estimated contributions to the CO profile

35 error budget from the following uncertainties. There are, of course, possible sources of error in any stage of the instrumentation, and the parameters discussed here are considered to represent the dominant uncertainties.

The statistical noise,  $\Delta T$ , on a spectrum is governed by the so-called radiometer equation  $\Delta T = T_{sys}/\sqrt{\nu\tau}$ , with  $T_{sys}$  as the system noise temperature of the instrument,  $\Delta v$  as the channel bandwidth (Table 1), and  $\tau$  as the integration time for a measurement of the atmospheric signal. The uncertainty for the temperature profile used in an inversion is the same uncertainty used in Hoffman et al. (2011): 5% below 80 km, 10% above 100 km, and linearly interpolated in

- 5 between. Uncertainties in the spectroscopic parameters of the CO line are from the HITRAN 2008 catalogue and are: 1% for the line intensity, 2% for the air broadening parameter, and 5% for the temperature dependence of air broadening. Uncertainty in the line position is ignored as the frequency grid can be adjusted to match the line centre in a measurement, and the parameters associated with self-broadening are considered as having negligible impact due to the relatively low concentration of the observed gas (Ryan and Walker, 2015). For the blackbody targets used in
- 10 calibration of the atmospheric measurement, an uncertainty of 2 K is assumed in the temperature of the hot and cold target: a conservative estimate that accounts for fluctuations and drifts in temperature. The uncertainty in the pointing of the instrument to the sky is estimated as 1°. This is an order of magnitude higher than the precision of the motor that controls pointing, to allow for a possible offset in the orientation of KIMRA.
- The resulting error estimates as well as a sum, in quadrature, of the errors are plotted in Figure 2, and show that the predominant error in the CO profiles comes from the statistical measurement noise on the spectrum. This peaks at 30% of the mean KIMRA CO profile, at 53 km altitude. The error from uncertainties in the temperature profile also shows a significant contribution to the total error in the retrievable altitude range above approximately 60 km. The error due to noise on the spectrum is calculated during each inversion and is provided in the supplemental data with the corresponding CO profiles. The error profile in Figure 2 is an average over all measurements. The other plotted errors are calculated about the a priori CO profile and serve as an estimate of the respective error contributions to each measurement. To calculate these errors for individual measurements is computationally expensive and so the values plotted in Figure 2 are provided in the supplemental data.

**2.5 Smoothing error and interpretation of the KIMRA profiles**

- Smoothing error in the profiles arises from the limited vertical resolution of the retrieved profile and can be calculated with the OEM using the averaging kernels and a priori information for a profile. Smoothing error can be large for ground-based profiling instruments due the small altitude spacing between retrieval grid points, chosen here for numerical stability in the inversion (Eriksson, 1999), compared to the actual vertical resolution of the retrieved profiles (see Figure 1 (c) and (d)). Smoothing error should be assessed when one wishes to use or interpret the CO profiles without consideration of the accompanying averaging kernels. As this is not a recommended use of the data, the
- 30 smoothing error is not assessed here. If using KIMRA profiles to say something of the absolute value of CO at a given retrieval altitude, one must be aware that a CO value at a retrieval grid point contains information from a range of altitudes with a sensitivity governed by the shape of the corresponding averaging kernel. Smoothing error can be accounted for when comparing KIMRA CO profiles to those of an instrument/model that has a significantly different vertical resolution by using the KIMRA averaging kernel matrices and a priori to smooth
- 35 (Rodgers and Connor, 2013) the other profiles so that they have a similar vertical resolution. KIMRA profiles can be used to observe changes in CO concentrations over time, provided there is not a significant difference in the averaging

kernels (and thus measurement response) of the profiles over that time. In particular, care should be taken with data near the edges of the retrievable altitude range (where the measurement response is decreasing towards 0.8), as the measurement response in this region can change quickly when there are sharp changes in atmospheric CO.

**3** Comparison with MLS**

This section presents a comparison of CO profiles from KIMRA and MLS that are colocated in space and time. A description of the MLS instrument is given in Walters et al. (2006). Version 4.2 of the CO data (Schwartz et al., 2015) is used here, a description of which can be found in Livesey et al. (2015). Version 4.2 CO data covers the pressure range of 215-0.0046 hPa. The precision of the CO profile reaches a maximum (largest) value of 1.1 ppmv at the highest retrieval layer (0.0046 hPa). In the middle atmosphere the dataset has a positive bias of approximately 20% compared with the ACE-FTS satellite instrument. This estimate is given by Livesey et al. (2015) using a validation of the MLS version 2.2 CO data (Pumphrey et al., 2007), which showed a positive bias of approximately 30%: Later

versions than 2.2 show slight lowering of the MLS values, bringing them closer to the ACE-FTS data.

**3.1 Colocation of KIMRA and MLS measurements**

For a given KIMRA measurement, a coincident MLS measurement was defined as follows. MLS measurements that
were made within 4 hours, ±2° latitude and ±10° longitude of the KIMRA measurement, and lie in the same position relative to the vortex edge as the KIMRA measurement (inside, outside, or within the edge of the polar vortex) were identified. The MLS measurement from this group that was closest, along a great circle, to the KIMRA measurement was chosen as coincident. A given KIMRA/MLS measurement could only be considered coincident with one MLS/KIMRA measurement. The location of a measurement with respect to the polar vortex was determined using scaled potential vorticity (sPV) values from NASA's Global Modeling and Assimilation Office's (GMAO's) MERRA

- (Modern Era Retrospective analysis for Research and Applications) (Rienecker et al., 2011). The sPV values for KIMRA were calculated geometrically along the instrument's line of sight. sPV values of 1.6 and 1.2  $\times 10^{-4}$  s-1 have been used extensively in previous works (e.g. Manney et al., 1994, 2007, 2011; Jin et al., 2006) to define the respective inner and outer edge of the vortex, and the same values are used here.
- 25 The location of a measurement, its position relative to the vortex, and the distance between measurements were calculated at 50 km altitude. The position relative to the vortex changes with altitude, which means that a single profile may simultaneously contain information from inside/outside/the edge of the polar vortex. 50 km was chosen as the altitude to define the three positions relative to the vortex because it divides the CO measurements from KIMRA into the three most distinct populations of concentration values. This was tested by using different altitudes to define the
- 30 position relative to the vortex, calculating partial (46-66 km and 66-86km) CO column concentrations, and testing whether the column concentrations were significantly different from each other when grouped by vortex relative position. This is not to say that that there should always be three distinct air masses of CO but it can be expected based on the strong cross-vortex gradient of the gas (see Section 1). The strong CO gradient during winter has recently been used in a chemical definition of the mesospheric vortex (Harvey et al., 2015).

Figure 3 shows the results for the KIMRA CO columns with vortex relative positions calculated using sPV values at 40, 50, and 60 km. The sPV information is available up to approximately 62 km. Using 40 km, the partial columns defined as outside and in the edge of the vortex are statistically indistinguishable at the 5% significance level using an unpaired two-sample t-test, and using 60 km, the inside and edge of vortex 46-66 km partial columns are

- 5 indistinguishable. With the same test and using 50 km, the three groups of partial columns comprise three distinct populations of concentration values in both the stratosphere and the mesosphere. Using other altitudes to define vortex relative position also produced distinguishable concentrations and 50 km was chosen as it gives the most distinct vortex relative concentration values in both the stratosphere and mesosphere, and also because the sPV-defined location of the vortex edge in the upper stratosphere becomes much less well-defined as the winter progresses (Manney
- 10

**et al., 1997). Both the sPV and CO concentration gradients will be less distinct before/after and during the formation/breakdown of the polar vortex.**

**3.2 Comparison of colocated measurements**

There are 916 coincident profiles found using the criteria in the previous section. The MLS CO profiles have a vertical resolution more than twice as fine as that of the KIMRA profiles: 3.5 - 5 km from the upper troposphere to the lower mesosphere, and 6 - 7 km in the upper mesosphere (Livesey et al., 2015). Because of this, the MLS profiles were smoothed with the averaging kernels of coincident KIMRA profiles to account for the difference in vertical resolution. MLS CO profiles are retrieved up to 0.001 hPa in the atmosphere and use a constant CO concentration above this. Because there is some low sensitivity of the KIMRA CO profiles to atmospheric concentrations above this (see the averaging kernels in Figure 1), the MLS profiles were extrapolated from 0.001 hPa before smoothing. A linear

- 20 extrapolation in pressure space was used to extend the MLS profiles instead of using scaled KIMRA a priori information so as to avoid creating artificial agreement between KIMRA and MLS. Figure 4 shows the mean of the extrapolated and the original MLS profiles, as well as the KIMRA a priori for comparison. The extrapolated profile is considered a more realistic representation of the atmosphere than the constant value provided for MLS at these altitudes. MLS values at 82 km and 84 km are often defined as unusable for scientific work due to a too-high a priori
- 25 contribution (Livesey et al., 2015) and the precision is given as negative in this case. The CO values are still considered here as useful for comparative purposes, but quantities derived using the precision (e.g., the slope of a line) are not meaningful at these altitudes. Figure 5 (a-e) shows the results of the comparison of the 916 pairs of coincident CO profiles found between January 2011 and May 2014. The mean of KIMRA and MLS CO profiles are shown (a) as well as the mean of the differences (bias) in the coincident profiles in ppmv (b) and relative to the mean of KIMRA
- 30 and MLS profiles (c), the sample Pearson correlation coefficient for the profiles at each altitude (d), and the slope of a line of best fit to KIMRA vs. MLS at each altitude layer (e), explained below. The times between coincident measurements and the location of the coincident MLS profiles are also shown (f). The statistics in Figure 5 are calculated using all colocated pairs of profiles and are also assumed to be representative of subsets of the data as there appears to be no particular time of the year during which the differences are more/less pronounced.
- The mean profiles show a maximum absolute bias of ~0.65 ppmv at 68 km, and a maximum relative bias of 22% (0.44 ppmv) at 60 km, with bias being defined as KIMRA minus MLS. The standard deviation of the differences in

the profiles peaks at 21% at 60 km. These standard deviation values are similar in magnitude to the estimated uncertainties in the KIMRA CO profiles (Figure 2). The standard error on the bias is not shown in Figure 5 (b) and (c) because it is very small due to the number of coincidences, but rather the standard deviation of the differences is shown as the whiskers on the bias. The combination defines a one-sigma space in which a KIMRA profile lies with

- 5 respect to an MLS profile. The correlation of the profiles is high at all altitudes, only dropping below 0.90 above 82 km. The correlation between KIMRA and unsmoothed MLS profiles is also plotted, with values between 0.81 and 0.90. Previous CO retrievals for winter 08/09 and 09/10 (Hoffmann et al., 2011) showed a bias with respect to MLS, increasing with altitude, with a value of > 5 ppm at 80 km. The structure of this bias was also shown in comparisons of KIMRA with ACE-FTS and MIPAS, and it appears that this attribute is not present in the retrievals presented here.
- 10 The slope of a line of best fit to KIMRA vs. MLS measurements was calculated individually for each altitude layer as follows: for a given retrieval grid point the slope and intercept (or regression coefficients) for a line of best fit to the KIMRA and MLS values was calculated, accounting for errors in the abscissa (MLS) and ordinate (KIMRA) values according to York et al. (2004). Two cases of KIMRA CO error estimates were used when calculating a line of best fit: the first being the measurement error in the profile (the error due to statistical noise on the spectrum (Section 2.4)),
- 15 and the second being twice the measurement error. The former is an underestimation of the true error on the profile as there are more error sources than measurement error alone, and the latter is likely an overestimation of the true error as the measurement error is a predominant source of error in the profile (Figure 2). The results (Figure 5 (e)) show that the slope is always greater than the ideal value of 1.0 for both KIMRA error estimates, meaning that KIMRA shows a greater range of CO concentrations at all altitudes and the variation is not explained by the estimated random
- 20 errors in the profile. The reason for the difference could be due to errors, e.g. spectroscopic information, calibration, baseline wave signatures, that can have a contribution that is neither truly systematic nor random. The difference in calculated slopes for the two KIMRA error estimates is insignificant within the standard error, likely due to the large natural variation in CO concentrations. Despite the > 1.0 slope values, KIMRA and MLS are considered to show agreement, according to the level of difference and correlation between profiles.

**25 4 Extension of the KIMRA dataset**

After establishing a reliable inversion scheme through comparison with MLS, the KIMRA dataset is extended in time by substituting ECMWF Operational Analyses model output for the SSMIS temperature data in the inversion (see Section 2.2).

**4.1 The effect of temperature input on KIMRA CO profiles**

30 ECMWF temperatures are available four times per day, in six-hourly intervals beginning at midnight. The same filtering procedure for the retrieved data is employed as outlined in section 2.3. To evaluate the effect of using a different temperature dataset as input to the inversion, the two KIMRA datasets are compared where they overlap between January 2011 to May 2014 and the results are shown in Figures 6 and 7.

A comparison of the two sets of temperature profiles (ECMWF/MSIS minus/vs. NCEP/SSMIS/MSIS) is shown in Figure 6. (a) and (b) show the mean and standard deviation of the differences between temperature profiles in absolute and relative units, (c) shows the correlation at each altitude, and (d) shows the slope of the lines of best fit at each altitude. The same is shown for the two respective sets of CO profiles in Figure 7. No smoothing with averaging kernels is applied to the data.

- The bias for the temperature profiles is very low below 50 km, showing good agreement between ECMWF and NCEP output, as well as the lower altitude SSMIS data, and then moves to a minimum of ~ -4% at 68 km. The maximum in the bias is about 4% at 118 km. The correlation is high below 50 km but has minima of < 0.50 at ~ 70 km and ~ 0.80 at 100 km. It should be noted that while MSIS is used in both temperature datasets, the time of the MSIS output is governed by the times for the ECMWF output and the SSMIS measurements, and so the high-altitude (> 0.01 hPa) temperature values are not necessarily equal for the two inversion setups. The slopes of the lines of best fit were
- 10 calculated using the same temperature error estimate, as described in Section 2.4, for each dataset. The slope is within 11% of 1.0 below 50 km altitude, above which it decreases to around 0.65 at 56 km, before increasing to 1.4 at 66 km, and then varies about 1 with another peak of 1.3 at 102 km.

There is a general positive bias in the CO profiles that use ECMWF/MSIS, seen in (a) and (b) of Figure 7. The bias is small, reaching a maximum of  $\sim$  5% in the range of 68-78 km. The correlation of the profiles is very high, greater than

15 0.98 at all altitudes below 82.5 km. The slopes of the lines of best fit were calculated with the same error estimates described in Section 3.2. A value of 1.0 lies in the range of standard error of the slope below 56 km and above 80 km, and between these altitudes reaches a maximum of 1.06. As the only difference in the inversion setups is the temperature input, it follows that any inequalities of the respective KIMRA CO profiles are ultimately due to this difference. Overall, the CO profiles using the differing temperature inputs shown here agree very well.

**20 4.2 KIMRA CO dataset from 2008 to 2015**

Figure 8 shows the KIMRA CO dataset between December 2008 and May 2015. Daily averaged CO concentrations between 46 and 86 km are shown. Data gaps in this time range are due to non-operation of the instrument or a lack of CO spectral line measurements. Data from winter 2015/2016 are unavailable due to a failure of the KIMRA cooling system.

- 25 While it is impossible to fully characterise the concentrations shown without inclusion of other instrument data and/or model output, some observations are made here. The beginning of each winter (from about September through November) shows a movement of CO to lower altitudes, which can in general be expected as predominantly due to vertical advection (Allen et al., 1999; Minschwaner et al.,2010; Solomon et al., 1985). The high CO concentrations remain for most of winter before decreasing again from March onwards, generally due to loss of CO because of
- 30 increased ·OH and movement of low-CO air from lower latitudes as the final warming of the pole occurs. Signatures of "major" SSWs (during which the 10 hPa zonal circulation becomes easterly at 60° N), beginning 24th January, 26th January, and 6th January, 2009, 2010, and 2013, respectively, can be seen by the quickly (order of days) decreasing CO concentrations around these dates, and then the subsequent increases as the vortex recovered (see Section 1) (Manney et al., 2009, 2015). The effects of a "minor" SSW (during which the 10 hPa zonal circulation remains
- 35 westerly at 60° N), in early January 2015 (Manney et al., 2015) can also be seen. Decreases in CO concentrations during SSWs are mainly due to the influx of lower latitude air as the polar vortex destabilises. There are other visible

fluctuations in the presented KIMRA CO data over various timescales which, while not interpreted here, can be used in the characterisation of winter-time dynamics above Kiruna.

**5. Data availability**

The current KIMRA CO dataset (between December 2008 and May 2015), can be accessed publically through
PANGAEA Data Publisher for Earth and Environmental Science at: https://doi.pangaea.de/10.1594/PANGAEA.861730. Metadata is also provided including the averaging kernel matrices for each measurement. It is recommended to use the averaging kernels specific to each CO profile when using the data in a comparison with model output or a dataset with significantly different altitude resolution.

**6. Conclusion**

- 10 The aim of this work was to create a self-consistent dataset of strato-mesospheric CO profiles above Kiruna, using measurements from the ground-based microwave radiometer, KIMRA, and an optimal estimation technique. The resulting profiles cover an average altitude range of 48-84 km. The retrievable altitude limits vary with the SNR of each measurement, and CO VMRs above 70 km should be treated with care due an offset in centres of the corresponding averaging kernels above this altitude. As a test of the validity of the KIMRA observing system and to
- 15 create a reference dataset, CO was first retrieved using atmospheric temperature information from the SSMIS satellite instrument, available within 2011-2014, and compared to CO data from MLS. The instruments show agreement, with KIMRA showing a maximum bias of ~ 0.65 ppmv at 68 km (corresponding to 14.7% of the mean CO value at 68 km), and a maximum relative bias of 22% (0.44 ppmv) at 60 km. Correlations with MLS are greater than 0.90 at most altitudes. KIMRA shows a larger range of atmospheric CO concentrations, compared to MLS, that is not explained
- 20 by the estimates of random errors in the data, and may be due to some combination of random and systematic uncertainty. Some differences in the KIMRA and MLS data can be expected due to imperfect colocation of the measurements and because MLS has a horizontal resolution of 200-300 km in the mesosphere and stratosphere: an order of magnitude wider than the beamwidth of KIMRA at these altitudes. The KIMRA dataset is extended in time (2008 2015) by substituting ECMWF Operational Analyses temperature output in place of the SSMIS data. The
- 25 extended KIMRA dataset shows a difference (bias) of less than 5% compared to the reference dataset (the one using SSMIS data), and correlations between the two are greater than 0.98 at most altitudes. There is a larger range (≤ 6%) of concentrations seen by the extended dataset, compared to the reference dataset over the same time period. The extended dataset currently spans the time between December 2008 to May 2015, with data gaps. Measurements are ongoing at IRF Kiruna.

**30 Author contribution**

[revised manuscript text omitted]

Nedoluha, G. E., Gomez, R. M., Neal, H., Lambert, A., Hurst, D., Boone, C., and Stiller, G.: Validation of long-term measurements of water vapor from the midstratosphere to the mesosphere at two Network for the Detection of Atmospheric Composition Change sites, J. Geophys. Res. Atmos., 118, 934–942, doi:10.1029/2012JD018900, 2013. Ohyama, H., Nagahama, T., Mizuno, A., Nakane H., and Ogawa, H.: Observations of stratospheric and mesospheric

25  $O_3$  with a millimeter-wave radiometer at Rikubetsu, Japan, Earth Planets Space, 68:34, doi:10.1186/s40623-016-0406-4, 2016,

Picone, J.M., Hedin, A.E., Drob, D.P., and Aikin, A.C.: NRL-MSISE-00 Empirical Model of the Atmosphere: Statistical Comparisons and Scientific Issues," J. Geophys. Res., doi:10.1029/2002JA009430, 2002.

Rogers, C. D. and Connor, B. J.: Intercomparison of remote sounding instruments, J. Geophys. Res., 108, 4116, doi:10.1029/2002JD002299, 2003.

Rogers, C. D.: Inverse methods for atmospheric remote sounding: Theory and practice. Vol 2. Series on atmospheric and ocean physics. Singapore: World Scientific, 2000.

Rosenkranz, P.W.:Water vapor microwave continuum absorption: A comparison of measurements and models, Radio Science, 33, 919–928, 1998.

35 Rosenkranz, P.W.: Absorption of microwaves by atmospheric gases, in: Janssen M.A.: Atmospheric remote sensing by microwave radiometry, Wiley, New York, 37-90, 1993.

Rothman, L. S., Gordon, I. E., Barbe, A., Chris Benner, D., Bernath, P. F., Birk, M., Boudon, V., Brown, L. R., Campargue, A., Champion, J. –P., Chance, K., Couderti, L. H., Dana, V., Devi, V. M., Fally, S., Flaud, J. –M., Gamache, R. R., Goldman, A., Jacquemart, D., Kleiner, I., Lacome, N., Lafferty, W. J., Mandin, J. –Y., Massie, S. T., Mikhailenko, S. N., Miller, C. E., Moazzen-Ahmadi, N., Naumenko, O. V., Nikitin, A. V., Orphal, J., Perevalov, V.

I., Perrin, A., Predoi-Cross, A., Rinsland, C. P., Rotger, M., Simeckova, M., Smith M. A. H., Sung K., Tashkun, S. A., Tennyson, J., Toth, R. A., Vandaele, A. C., and Vander Auwera, J.: The HITRAN 2008 molecular spectroscopic database, J. Quant. Spec. Rad. Trans., 110(9-10), 533-572, 2009.
 Palm, M., Hoffmann, C. G., Golchert, S. H. W., and Notholt, J.: The ground-based MW radiometer OZORAM on Spitsbergen – description and status of stratospheric and mesospheric O3 measurements, Atmos. Meas. Tech., 3,

- Pumphrey, H. C., Filipiak, M. J., Livesey, N. J., Schwartz, M. J., Boone, C., Walker, K. A., Bernath, P., Ricaud, P.,
  Barret, B., Clerbaux, C., Jarnot, R. F., Manney, G. L., and Waters, J. W.: Validation of middle-atmosphere carbon monoxide retrievals from MLS on Aura, J. Geophys. Res., 112, D24S38, doi:10.1029/2007JD008723, 2007.
  Rienecker, M. M., Suarez, M. J., Gelaro, R., Todling, R., Bacmeister, J., Liu, E., Bosilovich, M. G., Schubert, S. D.,
- 15 Takacs, L., Kim, G.-K., Bloom, S., Chen, J., Collins, D., Conaty, A., da Silva, A., Gu, W., Joiner, J., Koster, R. D., Lucchesi, R., Molod, A., Owens, T., Pawson, S., Pegion, P., Redder, C. R., Reichle, R., Robertson, F. R., Ruddick, A. G., Sienkiewicz, M., and Woollen, J.: MERRA: NASA's Modern-Era Retrospective Analysis for Research and Applications, J. Climate, 24, 3624–3648, doi:10.1175/JCLI-D-11-00015.1, 2011. Ryan, N. J., and Walker, K. A.: The effect of spectroscopic parameter inaccuracies on ground-based millimeter wave
- 20 remote sensing of the atmosphere. JQSRT, 161, 50-59, doi:10.1016/j.jqsrt.2015.03.012, 2015. Schwartz, M., Pumphrey, H., Livesey, N. and Read, W.: MLS/Aura Level 2 Carbon Monoxide (CO) Mixing Ratio V004, version 004, Greenbelt, MD, USA, Goddard Earth Sciences Data and Information Services Center (GES DISC), Accessed January 2016 at 10.5067/AURA/MLS/DATA2005.
- Solomon, S., Garcia, R. R., Olivero, J. G., Bevilacqua, R. M., Schwarzt, P. R., Clancy, R. T., and Muhleman, D. O.:
  Photochemistry and Transport of Carbon Monoxide in the Middle Atmosphere, J. Atmos. Sci., 42(10), 1072-1083, 1985.

Straub, C., Espy, P., Hibbins, R. E., and Newnham, D. A.: Mesospheric CO above Troll station, Antarctica observed by a ground based microwave radiometer. Earth Syst. Sci. Data, 5, 199–208, doi:10.5194/essd-5-199-2013, 2013.

Swadely, S. D., Poe, G. A., Bell, W., Hong, Y., Kunkee, D. B., McDermid, I. S., Leblanc, T.: Analysis and
 Characterization of the SSMIS Upper Atmosphere Sounding Channel Measurements, IEEE T Geosci. Remote., 46, 4, 962-983, 2008.

Waters, J., Froidevaux, L., Harwood, R., Jarno, R., Pickett, H., Read, W., Siegel, P., Cofield, R., Filipiak, M., Flower,D., Holden, J., Lau, G., Livesey, N., Manney, G., Pumphrey, H., Santee, M., Wu, D., Cuddy, D., Lay, R., Loo, M.,Perun, V., Schwartz, M., Stek, P., Thurstans, R., Boyles, M., Chandra, S., Chavez, M., Chen, G.-S., Chudasama, B.,

35 Dodge, R., Fuller, R., Girard, M., Jiang, J., Jiang, Y., Knosp, B., LaBelle, R., Lam, J., Lee, K., Miller, D., Oswald, J., Patel, N., Pukala, D., Quintero, O., Scaff, D., Snyder, W., Tope, M., Wagner, P., and Walch, M.: The Earth Observing System Microwave Limb Sounder (EOSMLS) on the Aura satellite, IEEE T. Geosci. Remote, 44, 1075–1092, 2006.

10 1533–1545, 2010.

York, D., Evensen, N. M., Martinez, M. L., and Delgado JDB. Unified equations for the slope, intercept, and standard errors of the best straight line, Am. J. Phys., 72(3), 367-375, 2004.

| System noise temperature  | ~1800 K (single sideband)                    |
|---------------------------|----------------------------------------------|
| Detector                  | Schottky diode at ~25 K                      |
| Sideband filter           | Martin-Pupplett interferometer               |
| Standing wave suppression | Path length modulator                        |
| Hot/Cold calibration      | Blackbodies at ~195 K/~293 K                 |
| Spectrometer              | FFTS. bandwidth/resolution: 110 MHz/ 107 kHz |

 Table 1: More general details of KIMRA at a glance. Also see Section 2.1.